# Integrated intra- and intercellular signaling knowledge for multicellular omics analysis

Dénes Türei[1] [ID], Alberto Valdeolivas[1] [ID], Lejla Gul[2], Nicolàs Palacio-Escat[1,3,4] [ID], Michal Klein[5] [ID], Olga Ivanova[1] [ID], Márton Ölbei[2,6] [ID], Attila Gábor[1] [ID], Fabian Theis[5,7] [ID], Dezső Módos[2,6] [ID], Tamás Korcsmáros[2,6] [ID] & Julio Saez-Rodriguez[1,3,*] [ID]

## Abstract

Molecular knowledge of biological processes is a cornerstone in omics data analysis. Applied to single-cell data, such analyses provide mechanistic insights into individual cells and their interactions. However, knowledge of intercellular communication is scarce, scattered across resources, and not linked to intracellular processes. To address this gap, we combined over 100 resources covering interactions and roles of proteins in inter- and intracellular signaling, as well as transcriptional and post-transcriptional regulation. We added protein complex information and annotations on function, localization, and role in diseases for each protein. The resource is available for human, and via homology translation for mouse and rat. The data are accessible via *OmniPath*'s web service (https://omnipathdb.org/), a Cytoscape plug-in, and packages in R/Bioconductor and Python, providing access options for computational and experimental scientists. We created workflows with tutorials to facilitate the analysis of cell–cell interactions and affected downstream intracellular signaling processes. *OmniPath* provides a single access point to knowledge spanning intra- and intercellular processes for data analysis, as we demonstrate in applications studying SARS-CoV-2 infection and ulcerative colitis.

**Keywords** intercellular signaling; ligand-receptor interactions; omics integration; pathways; signaling network

**Subject Categories** Computational Biology; Methods & Resources; Signal Transduction

**Mol Syst Biol. (2021) 17: e9923**

## Introduction

Cells process information by physical interactions of molecules. These interactions are organized into an ensemble of signaling pathways that are often represented as a network. This network determines the response of the cell under different physiological and disease conditions. In multicellular organisms, the behavior of each cell is regulated by higher levels of organization: the tissue and, ultimately, the organism. In tissues, multiple cells communicate to coordinate their behavior to maintain homeostasis. For example, cells may produce and sense extracellular matrix (ECM), and release enzymes acting on the ECM as well as ligands. These ligands are detected by receptors in the same or different cells, that in turn trigger intracellular pathways that control other processes, including the production of ligands and the physical binding to other cells. The totality of these processes mediates the intercellular communication in tissues. Thus, to understand physiological and pathological processes at the tissue level, we need to consider both the signaling pathways within each cell type as well as the communication between them.

Since the end of the nineties, databases have been collecting information about signaling pathways (Xenarios *et al*, 2000). These databases provide a unified source of information in formats that users can browse, retrieve, and process. Signaling databases have become essential tools in systems biology and to analyze omics data. A few resources provide ligand–receptor interactions (Kirouac *et al*, 2010; Fazekas *et al*, 2013; Ramilowski *et al*, 2015; Armstrong *et al*, 2019; Efremova *et al*, 2020). However, their coverage is limited, they do not include some key players of intercellular communication such as matrix proteins or extracellular enzymes, and they are not integrated with intracellular processes. This is increasingly important as new techniques allow us to measure data from single cells, enabling the analysis of inter- and intracellular signaling. For example, the recent *CellPhoneDB* (Efremova *et al*, 2020) and *ICELLNET* (Noël *et al*, 2021) tools provide computational methods to prioritize the most likely intercellular connections from single-cell transcriptomics data, and *NicheNet* (Browaeys *et al*, 2019) expands this to intracellular gene regulation. A comprehensive resource of inter- and intracellular signaling knowledge would enhance and expedite these analyses.

1  Faculty of Medicine and Heidelberg University Hospital, Institute of Computational Biomedicine, Heidelberg University, Heidelberg, Germany
2  Earlham Institute, Norwich, UK
3  Faculty of Medicine, Joint Research Centre for Computational Biomedicine (JRC-COMBINE), RWTH Aachen University, Aachen, Germany
4  Faculty of Biosciences, Heidelberg University, Heidelberg, Germany
5  Institute of Computational Biology, Helmholtz Zentrum München, Neuherberg, Germany
6  Quadram Institute Bioscience, Norwich, UK
7  Department of Mathematics, Technical University of Munich, Garching, Germany
   *Corresponding author. Tel: +49 6221 5451334; E-mail: julio.saez@uni-heidelberg.de

To effectively study multicellular communication, a resource should (i) classify proteins by their roles in intercellular communication, (ii) connect them by interactions from the widest possible range of resources, and (iii) integrate all this information in a transparent and customizable way, where the users can select the resources to evaluate their quality and features, and adapt them to their context and analyses. Prompted by the lack of comprehensive efforts addressing principle (i), we built a database on top of *OmniPath* (Türei *et al*, 2016), a resource which has already shown the benefits of principles (ii) and (iii). The first version of *OmniPath* focused on literature curated intracellular signaling pathways. It has been used in many computational projects and omics studies. For example, to model cell senescence from phosphoarray data (An *et al*, 2020), or as part of a computational pipeline to predict the effect of microbial proteins on human genes (Andrighetti *et al*, 2020), and a community effort to integrate knowledge about the COVID-19 disease mechanism (Ostaszewski *et al*, 2020). The new *OmniPath* extends its scope to intercellular communication and its integration with intracellular signaling, providing prior knowledge for modeling and analysis methods. It combines 103 resources to build an integrated database of molecular interactions, enzyme-PTM *(post-translational modification)* relationships, protein complexes and annotations about intercellular communication, and other functional attributes of proteins.

We demonstrate with two case studies that we provide a versatile resource for the analysis of single-cell and bulk omics data. Leveraging the intercellular communication knowledge in *OmniPath*, we present two examples where autocrine and paracrine signaling are key parts of pathomechanism. First, we studied the potential influence of ligands secreted in severe acute respiratory syndrome coronavirus 2 *(SARS-CoV-2)* infection on the inflammatory response through autocrine signaling. We identified signaling mechanisms that may lead to the dysregulated inflammatory and immune response shown in severe cases. Second, we examined the rewiring of cellular communication in *ulcerative colitis* (UC) based on single-cell data from the colon. By analyzing downstream signaling from the intercellular interactions, we found pathways associated with the regulatory T cells targeted by myofibroblasts in UC.

## Results

We used four major types of resources: (i) molecular *interactions*, (ii) *enzyme-PTM* relationships, (iii) protein *complexes*, and (iv) molecule *annotations* about function, localization, and other attributes (Fig 1A). The *pypath* Python package combined the resources from those four types to build four corresponding integrated databases. Using the *annotations*, *pypath* compiled a fifth database about the roles in intercellular communication (*intercell*; Fig 1B). The ensemble of these five databases is what we call *OmniPath*, combining data from 103 resources (Fig 1A and Dataset EV1).

### A focus on intercellular signaling

To create a database of intercellular communication, we defined the roles that proteins play in this process. Ligands and receptors are main players of intercellular communication. Many other kinds of molecules have a great impact on the behavior of the cells, such as matrix proteins and transporters (Fig 2A). We defined eight major (Fig 2) and 17 minor generic functional categories of intercellular signaling roles (Datasets EV6 and EV10). We also defined ten locational categories (e.g., *plasma membrane peripheral*), using in addition structural resources and prediction methods to annotate the transmembrane, secreted and peripheral membrane proteins. Furthermore, we provide 994 specific categories (e.g., *neurotrophin receptors*). Each generic category can be accessed by resource (e.g., *ligands from HGNC*) or as the combination of all contributing resources (Fig EV4). To provide highly curated annotations, we checked every entry in each category manually against UniProt datasheets to exclude wrong annotations. Overall we defined 1,170 categories and provided 54,330 functional annotations about intercellular communication roles of 5,781 proteins.

We collected the proteins for each intercellular communication functional category using data from 27 resources (Fig 2B, Dataset EV6). Combining them with molecular interaction networks from 48 resources (Dataset EV2), we created a corpus of putative intercellular communication pathways (Fig 2C). To have a high coverage on the intercellular molecular interactions, we also included ten resources focusing on ligand–receptor interactions (Figs 3 and EV1).

Many of the proteins in intercellular communication work as parts of complexes. We therefore built a comprehensive database of protein complexes and inferred their intercellular communication roles: a complex belongs to a category if and only if all members of the complex belong to it. We obtained 14,348 unique, directed transmitter–receiver (e.g., ligand–receptor) connections, around seven times more than the largest of the resources providing such kind of data. We also mapped a textbook table (Cameron & Kelvin, 2013) of 131 cytokine–receptor interactions to the ligand–receptor resources. As the textbook contains well-known interactions, many of the resources cover more than 90% of them (Fig 2D). This large coverage is achieved by not only integrating ten ligand–receptor resources, but also complementing these with data from annotation and interaction resources.

An essential feature of this novel resource is that it combines knowledge about intercellular and intracellular signaling (Table 1). Thus, using *OmniPath* one can, for example, easily analyze the intracellular pathways triggered by a given ligand or check the transcription factors (TFs) and microRNAs (miRNAs) regulating the expression of such ligands.

### *OmniPath*: an ensemble of five databases

The abovementioned intercellular database exists in *OmniPath* together with four further databases (Fig 1B), supporting an integrated analysis of inter- and intracellular signaling.

### *The network of molecular interactions*
The *network* database part covers four major domains of molecular signaling: (i) protein–protein interactions (PPI), (ii) transcriptional regulation of protein-coding genes, (iii) miRNA–mRNA interactions, and (iv) transcriptional regulation of miRNA genes (TF-miRNA). We further differentiated the PPI data into four subsets based on the interaction mechanisms and the types of supporting evidence: (i) literature curated activity flow (directed and signed; corresponds to the original release of *OmniPath*; Türei *et al*, 2016), (ii) activity flow

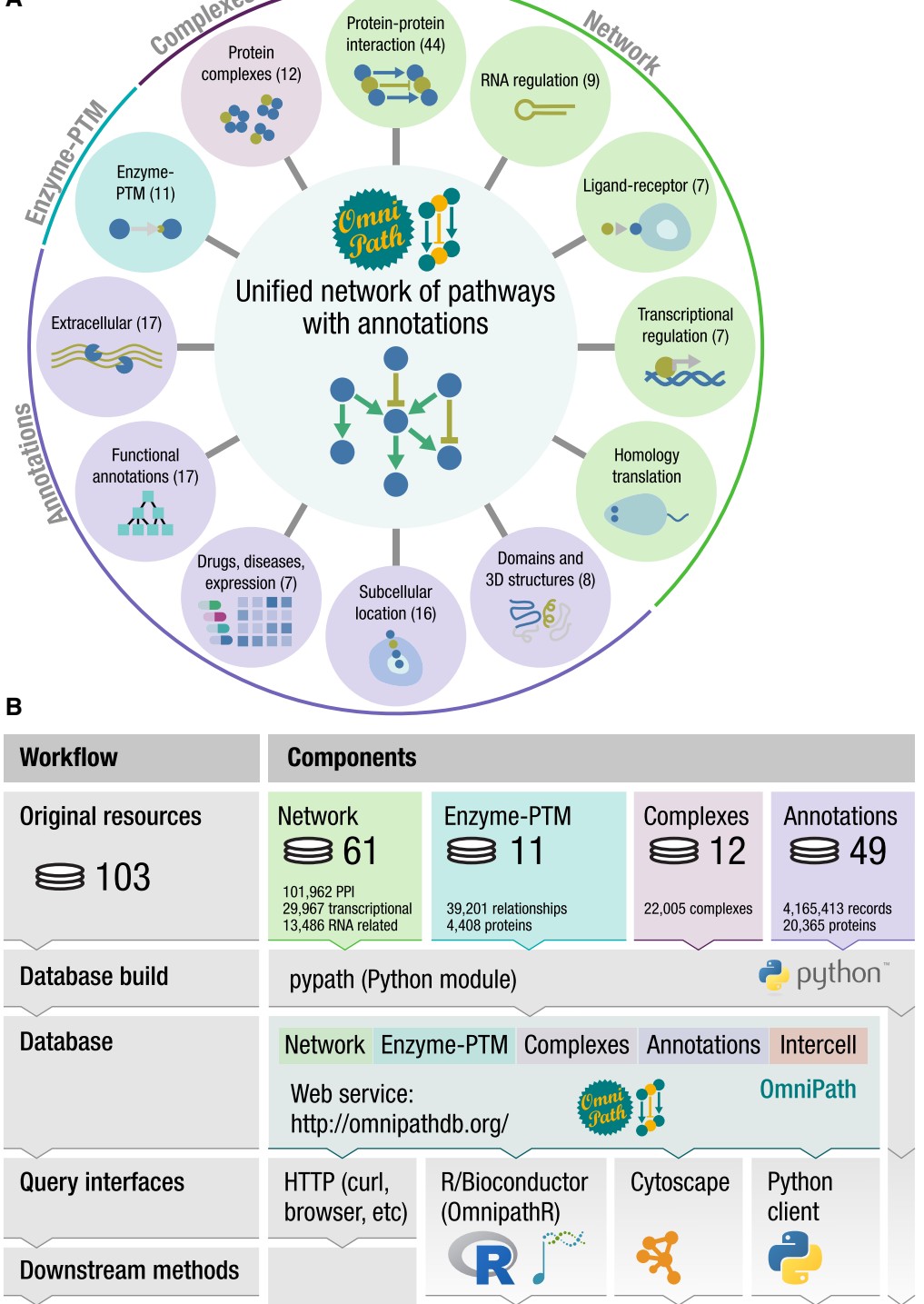

**Figure 1. The composition and workflow of OmniPath.**

A  Database contents with the respective number of resources in parentheses.

B  Workflow and design: OmniPath is based on four major types of resources, and the pypath Python package combines the resources to build five databases. The databases are available by the database builder software pypath, the web resource at https://omnipathdb.org/, the R package OmnipathR, the Python client omnipath, the Cytoscape plug-in and can be exported to formats such as Biological Expression Language (BEL).

with no literature references, (iii) enzyme–PTM, and (iv) ligand–receptor interactions (Fig 3A–C). Interaction data are extensively used for a variety of purposes: for building mechanistic models, deriving pathway and TF activities from transcriptomics data and graph-based analysis methods. In total, the resource contained

103,396 PPI interactions between 12,469 proteins from 38 original resources (Dataset EV2). The large number of unique interactions added by each resource underscores the importance of their integration (Figs EV1 and EV2, Appendix Fig S1). The interactions with effect signs, essential for mechanistic modeling, are provided by the

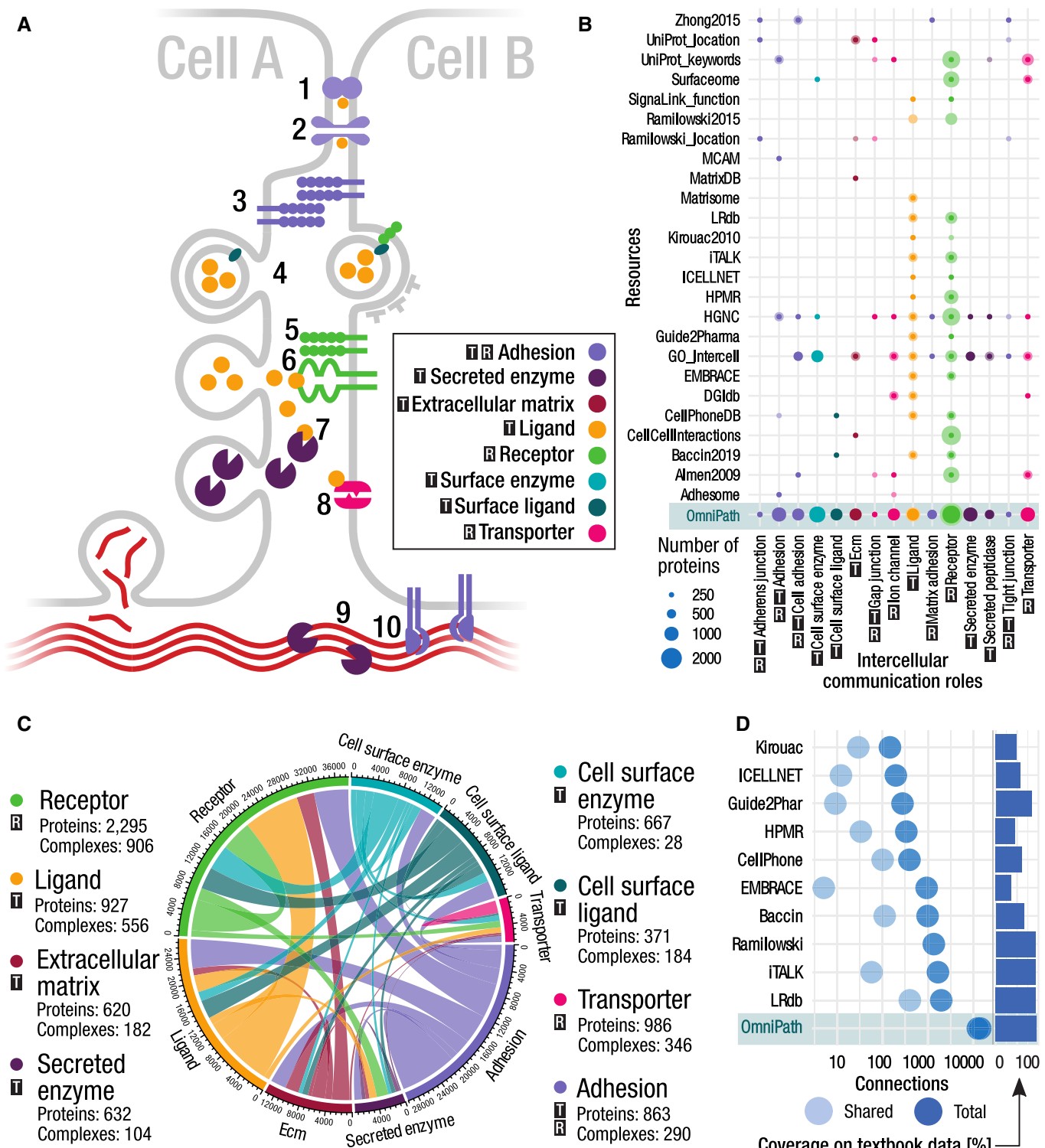

**Figure 2.**

**Figure 2.    The composition and representation of the intercellular signaling network.**

We assigned intercellular communication roles to proteins based on evidence from multiple resources. In all panels: T—transmitter; R—receiver.

A    Schematic illustration of the intercellular communication roles and their possible connections. Cells are physically connected by proteins forming tight junctions (1), gap junctions (2), and other adhesion proteins (3); they release vesicles which can be taken up by other cells (4); some receptors form complexes (5) to detect secreted ligands (6); transporters might also be affected by factors released by other cells (8); enzymes released into the extracellular space act on ligands and the extracellular matrix (7, 9); cells release the components of the extracellular matrix and bind to the matrix by adhesion proteins (10).

B    The main intercellular communication roles (x axis) and the major contributing resources (y axis). Size of the dots represents the number of proteins annotated to have a certain role in a given resource. The darker areas represent the overlaps (proteins annotated in more than one resource for the same role) while the lighter color denotes those unique to that resource.

C    The intercellular communication network. The circle segments represent the eight main intercellular communication roles. The edges are proportional to the number of interactions in the OmniPath PPI network connecting proteins of one role to the other.

D    Number of unique, directed transmitter–receiver (e.g., ligand–receptor) connections by resources. Bars on the right show the coverage of each resource on a textbook dataset of 131 well-known ligand–receptor interactions.

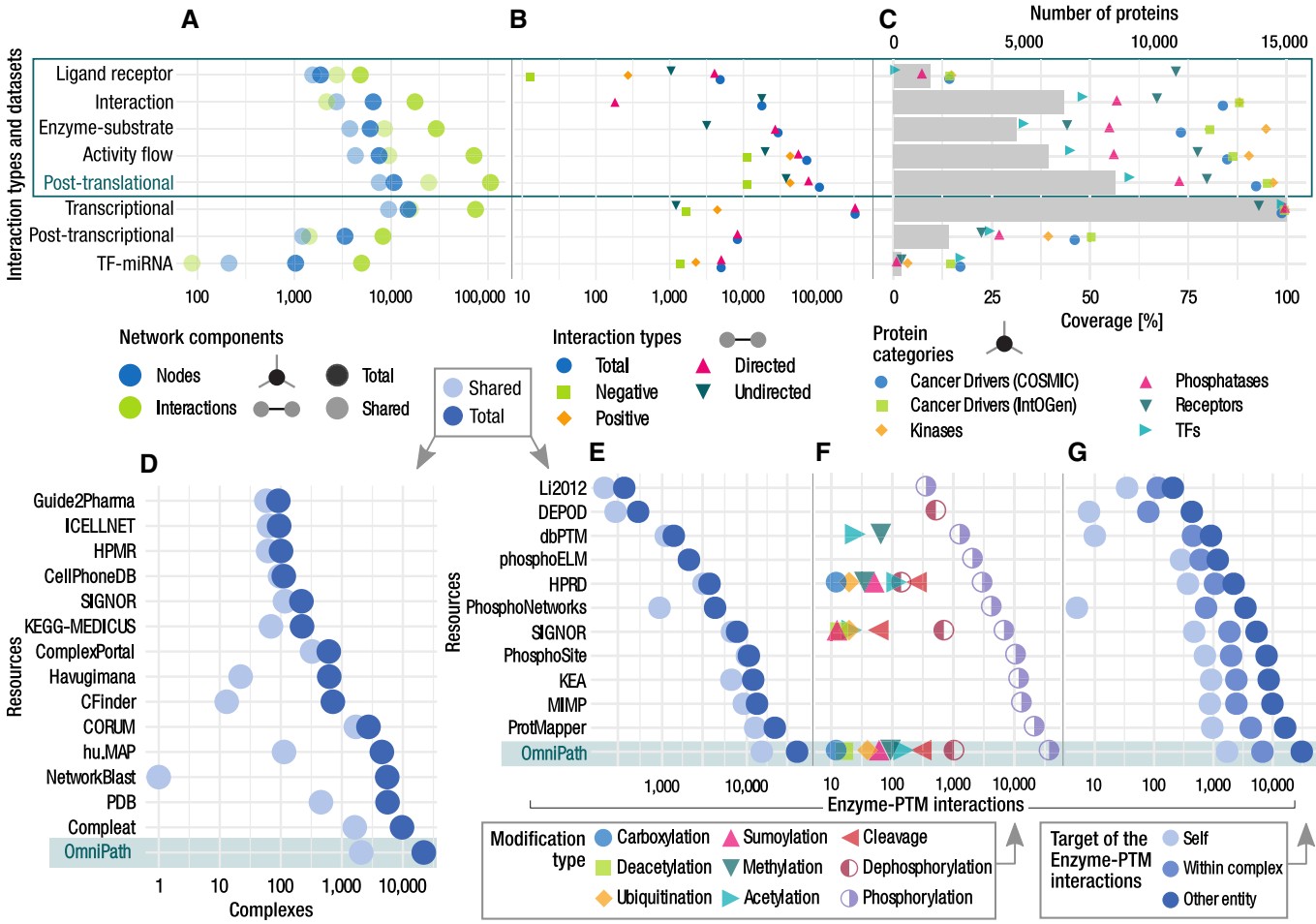

**Figure 3.    Quantitative description of the network, complex, and enzyme–PTM databases.**

A–C    Networks by interaction types and the network datasets within the PPI network. (A) Number of nodes and interactions. The light dots represent the shared nodes and edges (in more than one resource), while the dark ones show their total numbers. (B) Causality: number of connections by direction and effect sign. (C) Coverage of the networks on various groups of proteins. Dots show the percentage of proteins covered by network resources for the following groups: cancer driver genes from COSMIC and IntOGen, kinases from kinase.com, phosphatases from Phosphatome.net, receptors from the Human Plasma Membrane Receptome (HPMR) and transcription factors from the TF census. Gray bars show the number of proteins in the networks. The information for individual resources is in Figs EV1 and EV2, Appendix Fig S1.

D–G    On each panel, the bottom rows represent the combined complex and enzyme–PTM databases contained in OmniPath (D, E). Number of complexes (D) and enzyme–PTM (E) interactions by resource. (F) Enzyme–PTM relationships by PTM type. (G) Enzyme–PTM interactions by their target. Light, medium, and dark dots represent the number of enzyme–PTM relationships targeting the enzyme itself, another protein within the same molecular complex or an independent protein, respectively.

**Table 1.   Qualitative comparison of ligand–receptor and integrative databases.**

| Resource | Inter-actions | Directed inter-actions | Signs (positive/ negative) | Transcriptional regulation | Intracellular pathways | Intercellular communication roles | Protein complexes | Integrative resource | Literature curated |
|---|---|---|---|---|---|---|---|---|---|
| Baccin2019 (e) | yes | yes (a) | no | no | no | yes (f) | yes | yes | yes (g) |
| CellCellInteractions | yes | yes (a) | no | no | no | yes (l) | no | yes | no |
| CellPhoneDB | yes | yes (a) | no | no | no | yes (d) | yes | yes | yes |
| ConsensusPathDB | yes | no | no | yes | yes | no | no | yes | yes (g) |
| EMBRACE (e) | yes | yes (a) | no | no | no | yes | no | yes (k) | yes (g) |
| HPMR | yes | yes (a) | no | no | no | yes | no | no | yes |
| ICELLNET | yes | yes (a) | no | no | no | yes | yes | no | yes |
| iTALK (h) | yes | yes (a) | no | no | no | yes | no | yes | yes (g) |
| Kirouac2010 | yes | yes (a) | no | no | no | yes | no | no | yes |
| LRdb | yes | yes (a) | no | no | no | yes | no | yes | yes (g) |
| PathwayCommons | yes | yes (m) | no | yes | yes | no | yes | yes | yes (g) |
| Ramilowski2015 | yes | yes (a) | no | no | no | yes | no | yes | yes (g) |
| SignaLink | yes | yes | yes | yes (i) | yes | yes | no | yes (j) | yes (g) |
| OmniPath | yes | yes (b) | yes | yes | yes | yes (c) | yes | yes | yes (g) |

*OmniPath* combines resources to build a network with directions and effect signs, including intra- and intercellular signaling, transcriptional regulation, and annotates proteins as ligands or receptors. Here, we show which of these features are covered by other databases: those specialized in ligand–receptor interactions and two large integrative network databases (*ConsensusPathDB* and Pathway Commons). (a) Implicit: if we assume always the ligand affects the receptor; (b) As in some of the constituent resources the directions are implicit, certain directions in the combined network are implicit; (c) Provides not only ligand and receptor annotation but further categories, for example adhesion, transporter, ECM, etc; (d) Apart from secreted (mostly ligand) and receptor provides a few further categories: integrin, collagen, transmembrane, peripheral, etc; (e) Data are for mouse, homology translation is necessary to derive human data; (f) For ligands, provides certain classification, e.g., cytokine, ECM, secreted, etc; (g) Only in part is literature curated; (h) Ligand–receptor interactions are classified as growth factor, cytokine, checkpoint, or other; (i) Contains transcriptional regulation but that part is not integrated by OmniPath; (j) OmniPath only integrates its original literature curation, not the secondary resources; (k) Only builds on Ramilowski et al; (l) Besides ligand and receptor only ECM; (m) Directionality information might be extracted from BioPAX.

activity flow resources (Appendix 1; Fig 3B). The combined PPI network covered 53% of the human proteome (SwissProt), with an enrichment of kinases and cancer driver genes (Fig 3C). The transcriptional regulation data in *OmniPath* were obtained from *DoRothEA* (Garcia-Alonso et al, 2019), a comprehensive resource of TF regulons integrating data from 18 sources. In addition, six literature curated resources were directly integrated into *OmniPath* (Dataset EV8). The miRNA–mRNA and TF–miRNA interactions were integrated from five and two literature curated resources, with 6,213 and 1,803 interactions, respectively. Combining multiple resources not only increases the coverage, but also improves quality. It makes it possible to select higher confidence records based on the number of resources and references. Cross-checking the interaction directions and effect signs between resources reveal contradictory information which is either a sign of mistakes or reflects on limitations of our data representation (Appendix 1; Appendix Figs S4). Overall, we included 61 network resources in *OmniPath* (Dataset EV2). Furthermore, *pypath* provides access to additional resources, including the *Human Reference Interactome* (Luck et al, 2020), *ConsensusPathDB* (Kamburov et al, 2013), *Reactome* (Jassal et al, 2020), *ACSN* (Kuperstein et al, 2015), and *WikiPathways* (Slenter et al, 2018).

### Enzyme-PTM relationships

In enzyme–PTM relationships, enzymes (e.g., kinases) alter specific residues of their substrates, producing so-called post-translational modifications (PTM). Enzyme–PTM relationships are essential for deriving networks from phosphoproteomics data or estimating kinase activities. We combined 11 resources of enzyme–PTM relationships mostly covering phosphorylation (94% of all) and dephosphorylations (3%) (Fig 3F). Overall, we included 39,201 enzyme–PTM relationships, 1,821 enzymes targeting 16,467 PTM sites (Fig 3E–G). Besides phosphorylation and dephosphorylation, only proteolytic cleavage and acetylation account for more than one hundred interactions. Most of the databases curated only phosphorylation, and *DEPOD* (Damle & Köhn, 2019) exclusively dephosphorylation. Only *SIGNOR* (Licata et al, 2020) and *HPRD* (Keshava Prasad et al, 2009) contained a large number of other modifications (Fig 3F). 60% of the interactions were described by only one resource, and 92% of them by only one literature reference (Fig 3E). Self-modifications, e.g., autophosphorylation and modifications between members of the same complex comprised 4 and 18% of the interactions, respectively (Fig 3G).

### Protein complexes

Many proteins operate in complexes, for example, receptors often detect ligands in complexes. To facilitate analyses taking into consideration complexes, we added to *OmniPath* a comprehensive collection of 22,005 protein complexes described by 12 resources from 4,077 articles (Fig 3D). A complex is defined by its combination of unique members. 14% of them were homo-multimers, 54% had four or less unique components while 20% of them had 18 or more. 71% of the complexes had stoichiometry information.

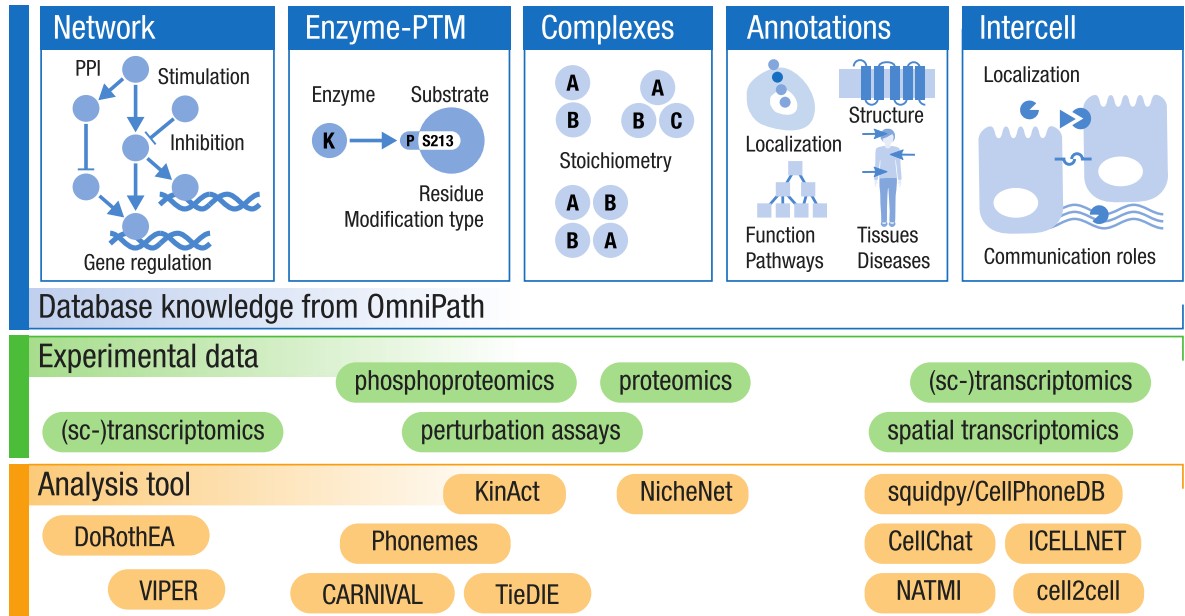

**Figure 4.**  **Examples of tools for omics data analysis that can be applied with the prior knowledge available in OmniPath**.

### Annotations: function, structure and localization

Annotations provide information about the function, structure, localization, classification, and other properties of molecules. We compiled the *annotations* database from 49 resources. The format of the records from each of these resources is different. The simplest ones only define a category of proteins, like *Cell Surface Protein Atlas (CSPA)* (Bausch-Fluck *et al*, 2015) that collects the proteins localized on the cell surface. More complex annotation records express a combination of multiple attributes. For example, each of the annotations from the *Cancer Pathway Association Database (CPAD)* (Li *et al*, 2020) contain seven attributes to describe a relationship between a protein or miRNA, a pathway, and their effect on a specific cancer type (Fig EV3). The pathway and gene sets are also part of the annotation database, as these are useful for functional characterization of omics data and enrichment analysis.

Overall, the *annotations* database included 5,475,532 records about 20,365 proteins, virtually the whole protein-coding genome, 19,566 complexes, and 182 miRNAs. The majority of the annotations for complexes were the result of our *in silico* inference: If all members of a complex share a certain annotation, we assign this annotation to the complex itself.

The *annotations* database can be used in different ways: Selecting one resource, its data can be reconstituted into a conventional data frame with attributes as columns and annotations as rows. Alternatively, specific sets of proteins can be queried, e.g., "the members of the *Notch pathway* according to *SIGNOR*" (Licata *et al*, 2020) or "the *hypoxia upregulated* genes according to *MSigDB*" (Subramanian *et al*, 2005). The annotations are helpful in omics data analysis; for example, they can be used for contextualization or enrichment analysis.

### Homology translation to rodents

*OmniPath* comprises human resources. We translated the network and the enzyme–PTM relationships to mouse and rat by protein homology using *NCBI HomoloGene*, covering 81 and 31% of the interactions for mouse and rat, respectively (Dataset EV9). In addition, *pypath* is able to translate to other organisms.

### Close connection to the analysis of omics data

The *OmniPath* databases are built by the *pypath* Python module and are distributed by the web service at https://omnipathdb.org/. We provide web service clients in R, Python, and Cytoscape (Ceccarelli *et al*, 2019). The clients not only query the OmniPath data but also offer convenient post-processing methods and integration with other software (Figs 1B and 4). The *OmnipathR* R client implements a full integration with *NicheNet*, a method for prioritizing ligands affecting cells based on transcriptomics data (Browaeys *et al*, 2019): A single *OmnipathR* function can be used to generate all inter- and intracellular knowledge required for NicheNet. The *omnipath* Python module, together with the single-cell data processing *scanpy* module (Wolf *et al*, 2018) and the *squidpy* reimplementation of the *CellPhoneDB* algorithm to infer ligand–receptor interactions between cell types (Efremova *et al*, 2020), provides an easy and efficient way to analyze intercellular communication. These applications and further examples are available as detailed tutorials at https://workflows.omnipathdb.org/. Here, a number of guides are available demonstrating various features of *OmniPath*, presenting the query parameters of the databases and showcasing downstream workflows.

### Case studies

*OmniPath* provides a single-access point to resources covering diverse types of knowledge. Thus, it can be used as an input to many analysis tools and is particularly useful for applications that span over molecular processes typically considered separately (Fig 4). To illustrate this, we used two examples where we extracted

from *OmniPath* different types of intra- and intercellular knowledge for computational analysis of bulk and single-cell RNA-Seq data.

### Analysis of intra- and intercellular processes in SARS-CoV-2-infected lung epithelial cancer cells

*NicheNet* is a recently developed method to prioritize ligand–target relationships between interacting cells by combining their expression data with prior knowledge on interaction networks (Browaeys *et al*, 2019). For this purpose, *NicheNet* explores the most consistent inter- and intracellular protein interactions in accordance with a given gene expression dataset. In the *NicheNet* publication, the authors collected different types of interactions from more than 20 databases to build a ligand–receptor network, a signaling network, and a gene regulatory network. Here, we built a network for analysis with *NicheNet* using exclusively *OmniPath*.

We used the *OmniPath* built network to investigate the mechanistic processes leading to the excessive inflammatory response and dysregulated adaptive host immune defense that may occur in severe *COVID-19* cases (Catanzaro *et al*, 2020). We studied the autocrine regulatory effect of ligands secreted in *SARS-CoV-2* infection of epithelial lung cancer cells *(Calu3;* Methods and Appendix 2; data from Blanco-Melo *et al*, 2020). Out of 117 ligands over-expressed in *SARS-CoV-2* infection, we selected for subsequent analysis the 12 best predictors of inflammatory response genes according to the distribution of correlation values (Fig EV5B) and *nichenetr* guidelines (Methods and Appendix 2).

Among them, we found various cytokines: interleukins (*IL23A* and *IL1A*), tumor necrosis factors (*TNF* and *TNFSF13B*), and chemokines (*CXCL5, CXCL9,* and *CXCL10*), known to be involved in the inflammatory response. NicheNet scores describing the potential influence of the 12 selected ligands on the set of inflammatory genes are significantly higher than on sets of randomly selected genes (average *P*-value = 3.25e-08 from Fisher's exact tests after 10 cross-validation rounds). Then, we explored the signaling events linking these ligands to their target genes (Fig 5A, Methods and Appendix 2). We identified several key proteins of the *JAK-STAT pathway* (JAK2, STAT1, STAT3, and STAT4), a main regulator of the inflammatory response, that has been suggested as a potential target to treat *COVID-19* (Bagca & Avci, 2020). We also found ligands that potentially trigger the *MAPK pathway* that has also been reported to be promoted by *SARS-CoV-2* infection (Bouhaddou *et al,* 2020; Treveil *et al,* 2021). To further characterize the potential medical relevance of these results, we investigated the drugs targeting the genes shown in Fig 5A (Dataset EV14). Among the most interesting results, we identified minocycline, an antibiotic, and anti-inflammatory drug targeting *CASP3* and *TNF*. Minocycline has been very recently proposed to alleviate the effects of SARS-CoV-2 severe infection in the central nervous system (Oliveira *et al,* 2020) (see extended results in Appendix 2).

In summary, we found mechanistic insights about inflammatory-related signaling cascades triggered by SARS-CoV-2 infection. The underlying interactions spanned different curated (and thus supported by literature) individual inter- and intracellular resources that we could leverage as they are all integrated in *OmniPath* (Fig 5A in Dataset EV13).

### Alteration of intercellular communication in ulcerative colitis

As a second case study, we used single-cell RNA-Seq data (Smillie *et al,* 2019) from *ulcerative colitis* (UC) to investigate paracrine signaling using *OmniPath*'s intra- and intercellular knowledge. UC is an inflammatory bowel disease (IBD) driven by an interplay of epithelial cells and resident mucosal immune cells. Hence, it would be desirable to investigate it with considering both cell type-specific intracellular signaling and cell–cell communication.

We limited our analysis to five cell types relevant in UC: dendritic cell (DC), macrophage, regulatory T cell (Treg), myofibroblast, and Goblet cell. We combined the cell type and condition-specific expression data with *OmniPath* to build intracellular and intercellular signaling networks (Appendix Fig S5). The total number of cell–cell connections was similar (Table EV1), while their identity and distribution were different between healthy and UC conditions. In healthy condition, all cell types were tightly connected to DCs while in UC to Treg cells (Fig 5B).

Using the *intercell* annotation database of *OmniPath*, we examined the type of intercellular interactions between these cell types. We found that in both healthy state and UC the ligand–receptor and adhesion connections were dominant and the cell junction type connections were less abundant in UC—which was expected due to the pathophysiology of the disease. Also in UC, we found a higher amount of ligand–receptor and adhesion connections between Treg cells and the other four cell types, supporting previously described alteration of Treg signaling in UC (Yamada *et al,* 2016).

To analyze the changes in Treg signaling more in detail, we combined the intercellular and intracellular databases from *OmniPath* and focused on the connection between myofibroblasts and Treg cells. The total number of intercellular connections are nearly the same in healthy and in UC conditions 472 and 478, respectively. However, the actual interacting proteins and their downstream effects are remarkably different (Fig 5C). This is mainly due to ligands from myofibroblasts or receptors on Treg cells expressed uniquely in one of the conditions. For example TGF-beta signaling is a known regulatory input of Treg cells (Wan & Flavell, 2008), and we found BMPR1A and ACVRL1, two receptors for the TGF-beta pathway, to be specific for healthy and UC conditions, respectively. Although there is no evidence for the role of ACVRL1 in Treg cells, the knockout of Bmpr1a contributes to gut inflammation (Shroyer & Wong, 2007). The changes in intercellular connections lead to major downstream signaling difference in Treg cells. To map the downstream effect, we built an intracellular network of Treg cells including two steps downstream of all recipient proteins targeted by myofibroblast effectors (Fig 5B). There were roughly two times more affected downstream proteins in Treg cells in UC than in healthy condition (835 versus 1,971), suggesting a wider regulatory impact of myofibroblasts on Treg cells. Using Reactome (Jassal *et al,* 2020) pathway enrichment analysis (Dataset EV11), we identified the main pathways in Treg cells affected differently by myofibroblasts in the two conditions. In healthy state, the MAPK, Toll-like receptor (TLR) 2/6, and TLR7/8 pathways were enriched that are known as key processes regulating immunosuppressive functions and suppressing the proinflammatory Th17 cells (Forward *et al,* 2010; Nyirenda *et al,* 2015; He *et al,* 2018). Meanwhile in UC, TLR4 and TLR3 pathways were affected by myofibroblasts, and these pathways are relevant in UC as they regulate inflammatory cytokine expression and decrease the abundance of Treg cells (Xiao *et al,* 2009; Cao *et al,* 2014).

Our analysis supports the fact that the normally anti-inflammatory effect of Treg cells in UC is deteriorated partially by myofibroblasts

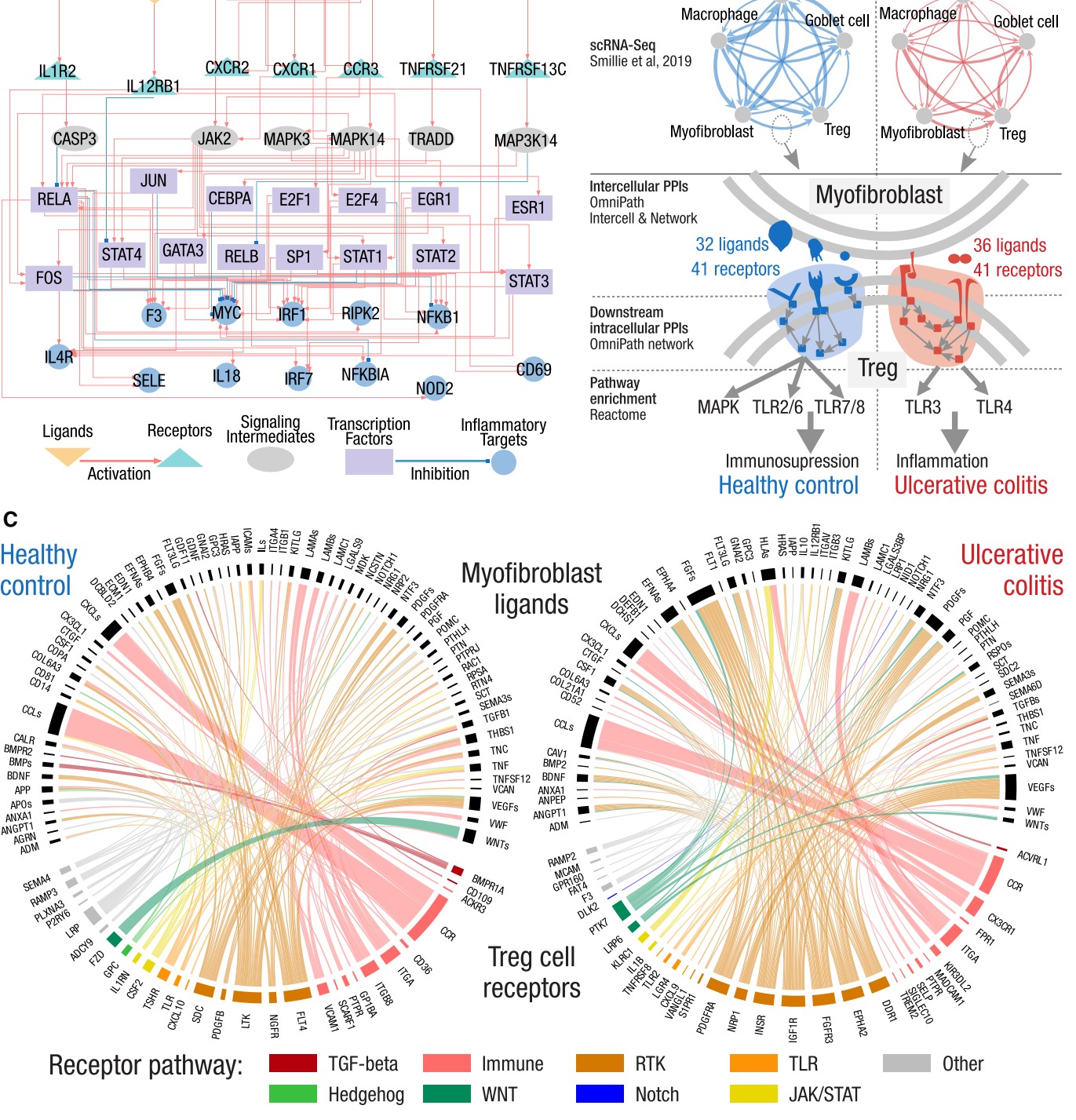

**Figure 5. Illustrations of the integrated analysis of inter- and intracellular signaling.**

A Inter- and intracellular signaling interactions linking the top predicted ligands over-expressed after SARS-CoV-2 infection to their potential immune response targets in the Calu3 cell line. Top ranked ligands (orange) connect to their potential receptors (turquoise) that trigger an intracellular cascade until reaching TFs (purple), that in turn regulate the expression of the target genes (blue). Signaling intermediates (gray) connect receptors to TFs across their shortest path.

B Intercellular connections and their downstream effect in UC compared with healthy control. Top: communication network of five cell types reconstructed from scRNA-Seq; the thickness of the edges is proportional to the number of intercellular connections. Bottom: condition-specific ligand–receptor connections between myofibroblasts and regulatory T cells trigger an immunosuppressive versus an inflammatory signaling in T cells, in healthy and UC, respectively.

C Condition-specific connections between myofibroblast ligands (upper semicircles, black) and Treg cell receptors (lower semicircles, colored by pathways) in ulcerative colitis (right) and healthy control (left). Pathway annotations from SignaLink. Immune—innate immune response, RTK—receptor tyrosine kinase, TLR—Toll-like receptor.

**Table 2.** Number of unique receptors and their first two neighbors in each comparison.

| | Pathway Commons network | | | | OmniPath network | |
|---|---|---|---|---|---|---|
| | Ramilowski annotations | | OmniPath annotations | | | |
| Conditions | Unique receptors | First two neighbors | Unique receptors | First two neighbors | Unique receptors | First two neighbors |
| Healthy | 2 receptors | 7553 proteins | 13 receptors | 9371 proteins | 36 receptors | 2476 proteins |
| Uninflamed UC | 2 receptors | 10138 proteins | 6 receptors | 11441 proteins | 41 receptors | 2879 proteins |

(Takahashi *et al*, 2006; West, 2019). We found key intercellular mechanisms leading to well-defined differential pathway activation profiles. This was achieved via our novel approach to: (i) determine cell–cell interactions both in a healthy and in disease states and (ii) map affected downstream intracellular signaling processes based on the proteins interacting between cells.

### Comparing Omnipath to other resources for cell–cell interaction analysis

The UC use case provided an opportunity to compare *OmniPath* against alternative sources of prior knowledge. We chose two widely used resources, *Pathway Commons (PC)* (Cerami *et al*, 2011) for network and *Ramilowski et al* (Ramilowski *et al*, 2015) for ligand–receptor annotations. Using the same workflow and expression data (Table 2), we investigated the myofibroblast–Treg cell interaction in three different network–annotation combinations (Appendix Fig S3): (i) *PC-Ramilowski*; (ii) *PC-OmniPath*; and (iii) *OmniPath-OmniPath* (i.e., as we presented in the use case).

Using the undirected *PC* network with the *Ramilowski* and the *OmniPath* annotation resulted in 523 and 3,136 ligand–receptor connections, respectively. The *OmniPath* PPI network with *OmniPath* annotations revealed 4,473 ligand–receptor connections indicating that this combination provides the largest coverage and more detailed data with directionality. In the intracellular network of Treg cells, using the *Ramilowski* annotation with *PC* network we found around 20 times less condition-specific receptors, compared with using *OmniPath* for both network and annotation, leading to a subsequent loss of downstream pathways in the former case. At the same time, the *PC* network with *OmniPath* annotations provided a large Treg cell downstream network containing ∼ 50% of the *PC* network, while using the *OmniPath* network resulted in a three times smaller network, covering ∼ 30% of the total OmniPath network. This is mainly due to the fact that *PC* provides a denser network than *OmniPath*, but undirected. Overall, *OmniPath* provides a high number of ligand–receptor interactions and directed interactions for downstream intracellular pathway analysis, complementing other meta-resources.

## Discussion

In the first version of *OmniPath* (Türei *et al*, 2016), we built a comprehensive knowledge of intracellular signaling pathways with the aim of providing prior knowledge for modeling methods. Here, we present a major redesign and extension of this resource, offering a single-access point to over 100 resources containing prior knowledge of not only intra- but also intercellular processes. To achieve this, we developed versatile annotations of intercellular communication roles, combined with a network covering intra- and intercellular

signaling as well as gene regulation. By defining the *transmitter*, *receiver*, and *mediator* roles, we layed out a new conceptual framework to describe intercellular communication and generalized the terms of *ligand* and *receptor* (Dataset EV10). This framework allows *OmniPath* to combine diverse resources in a uniform way. In *OmniPath*, the intercellular annotations and the network connections are independent from each other, achieving together a great flexibility. As intercellular communication becomes increasingly popular thanks to single-cell technologies, we believe that supporting it with database knowledge deserves a dedicated effort instead of doing *ad hoc* data integration within each study.

While integrative resources such as *STRING* (Szklarczyk *et al*, 2019), *PathwayCommons* (Cerami *et al*, 2011), *ConsensusPathDB* (Kamburov *et al*, 2013), *PathMe*, and *ComPath* (Domingo-Fernández *et al*, 2019) use mostly the major process description resources (e.g., *Reactome* (Jassal *et al*, 2020) and *ACSN* (Kuperstein *et al*, 2015)) and resources with undirected interactions (e.g., *IntAct* (Orchard *et al*, 2014) and *BioGRID* (Oughtred *et al*, 2019)), the *network* database of OmniPath focuses on activity flow representation, providing a convenient input for multiple analysis techniques (Touré *et al*, 2020); Appendix 1). *OmniPath* is not limited to literature curated interactions and it also includes activity flow, kinase–substrate, and ligand–receptor interactions without references as separate datasets, so that the users can decide which ones to use according to their purposes (Dataset EV2). The rich annotations allow users to dive into specific knowledge and extract information across resources. The knowledge in *OmniPath* is general in terms of cell type or physiological condition. In the process of data analysis and modeling, omics data help to make the database knowledge more context specific. As an alternative, one can use for example *Human Protein Atlas* (Uhlén *et al*, 2015) in the *OmniPath* annotations database to build tissue specific networks (https://workflows.omnipathdb.org/).

As we demonstrated here, *OmniPath* is able to deliver the input knowledge for different data analysis tools, such as *CellPhoneDB* (Efremova *et al*, 2020), *NicheNet* (Browaeys *et al*, 2019), *CellChat* (Jin *et al*, 2021), *ICELLNET* (Noël *et al*, 2021), *NATMI* (Hou *et al*, 2020), cell2cell (preprint: Armingol *et al*, 2020a), and *CARNIVAL* (Liu *et al*, 2019) to infer communication between (Armingol *et al*, 2020b) and within cell types. For some of the analysis tools, we provide dedicated software integration and workflows (https://workflows.omnipathdb.org/).

As our case studies illustrate, *OmniPath* can replace the tedious collection of information from many different databases. The first case study pointed to potential signaling mechanisms of autocrine origin in *SARS-CoV-2* infection which can contribute to the dysregulated inflammatory and immune response characteristic of severe *COVID* cases. Our study is limited to the relationship of autocrine signaling and inflammatory response and hence it does not cover the complete process of viral infection. In the second study, we illustrated

how conveniently *OmniPath* supports a combined analysis of inter- and intracellular signaling from single-cell transcriptomics data. While multiple studies mapped intracellular signaling pathways to intestinal tissue, only a few of them were able to do it in a cell type-specific manner using single-cell transcriptomics data (Smillie *et al,* 2019). Due to the lack of integrated resources, combined intra- and intercellular studies have been so far challenging and not standardized. This is currently a major bottleneck to understand better conditions such as gut inflammation, which is modulated by the interplay of epithelial cells and resident mucosal immune cells. The results of the case studies can guide designing co-culture experiments by prioritizing the most relevant cell types and pointing out the key cell–cell interaction types. For example, testing the role of CASP3 in the autocrine signaling we pointed out in the first study, and the specific ligand–receptor connections that altered the intestinal paracrine signaling in diseased condition in the second case study. In general, the outcome of Omni-Path-based analyses can define key candidates for more in depth investigations.

Over the past 4 years, we have kept developing *OmniPath,* adding new features and resources regularly. One of our main objectives for the future is to add more context information, e.g., cell type and physiological condition to the signaling network, and use scores to prioritize interactions and paths which contribute stronger to indirect causal relationships. Toward these aims, we plan to leverage text mining methods (Gyori *et al,* 2017; Kveler *et al,* 2018). We are also working on benchmarking the intercellular communication knowledge by deriving ground truth from experimental data (Armingol *et al,* 2020b). Furthermore, we envision to extend *OmniPath* with pathogen–host interactions (Treveil *et al,* 2021) and microbiome–host interactions (Andrighetti *et al,* 2020) in the near future.

In summary, we provide a new integrated resource of biological knowledge particularly valuable for network analysis and modeling of bulk and single-cell omics data. We anticipate that this knowledge will also be valuable to analyze the emergent spatially resolved omics data (Asp *et al,* 2020). To understand tissue architecture and function, it is crucial to study the spatial arrangement of the different cell types. Spatial transcriptomics technologies provide this information and hence help to prioritize the most likely ligand–receptor interactions. Fundamental questions about cell communication in tissues, such as how secreted ligands act on neighboring cells, can be addressed by analyzing spatially resolved data, combining data-driven (Sun *et al,* 2020; preprint: Tanevski *et al,* 2020) with prior knowledge-based (Browaeys *et al,* 2019; Liu *et al,* 2019; Efremova *et al,* 2020) approaches. *OmniPath* provides a framework to support these endeavors.

# Materials and Methods

## Terminology

In the manuscript, we use consistently the following three definitions to describe the structure of *OmniPath*:

- Database: collection of similar records in a uniform format integrated from multiple resources (network, enzyme-PTM, complexes, annotations, intercell).

- Dataset: a subset or variant of a database, e.g., the transcriptional interaction network is a dataset of the network database.
- Resource: any data source we use for building the databases.

## Database build

To build *OmniPath*, we developed a free software, the *pypath* Python module (https://github.com/saezlab/pypath, version 0.11.39). We built each segment of the database by the corresponding submodules and classes in *pypath*. In addition to the database building process, all modules rely on common utility modules from *pypath* such as the identifier translator or the downloading and caching service. *Pypath* downloads all data from the original sources. Many resources integrate data from other resources, we call these secondary resources and their relationships are listed in Dataset EV7.

### Network

For the *OmniPath* network, we converted the identifiers of the different molecules and merged their pairwise connections, preserving the literature references, the information about the direction, and effect sign (activation or inhibition).

In *OmniPath*, we included nine network datasets built from 61 resources (Dataset EV2). The first four datasets provide PPI ("*post_translational*" in the web service) while the others transcriptional and post-transcriptional regulation. At each point below, we highlight the label of the dataset in the web service.

1   We compiled the "omnipath" network as described in Türei et al (Türei *et al,* 2016). Briefly, we combined all resources we could get access to, that are literature curated and are either activity flow, enzyme–PTM, or undirected interaction resources. We also added network databases with high-throughput data. Then, we added further directions and effect signs from resources without literature references.
2   The "kinaseextra" network contains additional kinase–substrate interactions without literature references. The direction of these interactions points from the enzyme to the substrate.
3   In the "pathwayextra" network, we combined further activity flow resources without literature references. However, they are manually curated and many have effect signs.
4   In the "ligrecextra" network, we provide additional ligand–receptor interactions from large, comprehensive collections.
5   The "dorothea" network comes from DoRothEA database, a comprehensive resource of transcription factor–gene promoter interactions from literature curated databases, high-throughput experiments, binding motif and gene expression-based *in silico* inference, overall 18 resources(Garcia-Alonso *et al,* 2019). We included the interactions from DoRothEA subclassified by confidence levels from A to D, excluding the lowest confidence level E. In *OmniPath*, users are able to filter the TF–target interactions by confidence level.
6   Transcriptional regulation ("tf_target") directly from 6 literature curated resources. We show the size of the TF–target network at different settings in Dataset EV8.
7   In the "post_transcriptional" network, we combined 5 literature curated miRNA–mRNA interactions.

8   Transcriptional regulation of miRNA ("tf_mirna") from 2 literature curated resources.

9   lncRNA–mRNA interactions from 3 literature curated resources ("lncrna_mrna").

### Enzyme–PTM interactions

After translating the identifiers, we merged enzyme-PTM interactions from 11 databases (Dataset EV3) based on the identity of the enzyme, the substrate and the modified residue and its position. In addition, we discarded the records where the residue could not be found in any of the isoform sequences from UniProt (UniProt Consortium, 2019). For each enzyme–PTM interaction, we included the original sources and the literature references. We also kept the records without literature support, e.g., from high-throughput screenings or in silico prediction.

### Complexes

We combined 12 databases to build a comprehensive set of protein complexes (Dataset EV4). Seven of these databases provide information about the stoichiometry of the complexes while three contain only the lists of components. We translated the names of the components to UniProtKB accession numbers. We defined the complexes by their unique combination of members regardless of how the original resource processed the underlying experimental data. We merged the complexes based on their identical sets of components and preserved the stoichiometry if available. We represent each complex by the UniProt IDs of their components sorted alphabetically, separated by underscores and prefixed with "COMPLEX:". From the original sources, we kept the literature references, the human readable names (synonyms) and the PDB structure identifiers if available.

### Annotations

Annotation resources provide diverse information about the localization, function, or other characteristics of the molecules. We obtained annotations from 49 databases (Dataset EV5). For these databases, we translated IDs and extracted the fields with relevant information. Due to the heterogeneous nature of the data, in the annotation database, the content of the resources is not merged, but rather all entries are provided independently.

Each annotation record assigns one or more attributes to a molecule. One protein might have more than one annotation record from the same database. For example, Vesiclepedia (Pathan et al, 2019) provides two attributes: the vesicle type and the tissue where the protein has been detected. We combined the annotation resources into a uniform table where one column is the name of the attribute and the other is the value. As one record might have multiple attributes, the records are identified by unique numbers (Fig EV3). Providing the data in this format in our web service, it can be easily reconstituted to conventional tables with standard tools like tidyr (https://tidyr.tidyverse.org) in R or pandas (https://pandas.pydata.org) in Python.

### Complex annotations

Only four resources curate annotations of protein complexes; from these, we processed the complex annotations as we did for proteins. Furthermore, we inferred annotations for complexes based on the annotations of their components. We assigned the annotations to the complex if all components agreed in all attributes that we considered relevant, e.g., if all members of a complex belong to the WNT pathway then the complex is also annotated as a member of the WNT pathway.

### Intercellular signaling roles

From the resources used in annotations, we selected 26 with function, location, or structure information relevant in intercellular signaling. The relevant attributes we processed and combined to account for main roles in intercellular communication (e.g., ligand, receptor, ECM proteins) as well as the locational and topological properties (e.g., secreted, transmembrane). In addition, we built Boolean expressions from Gene Ontology terms to define the same categories. Overall we created 25 functional and 10 locational categories (Dataset EV6). Each category carries the attributes described in Dataset EV10 (Fig EV4). We manually checked the members of all the annotation groups, relying on literature knowledge and UniProt datasheets (UniProt Consortium, 2019), discarding the wrong annotations. We provide the classification of proteins and complexes by these categories in the intercell query of the web service.

### Identifier translation

For each type of molecule, we chose a reference database: for proteins the UniProtKB ACs while for miRNAs the miRBase (Kozomara et al, 2019) mature Acs. From these databases, we obtained a reference set of identifiers for each type of molecular entity and organism. We then used translation tables provided by them to map other kinds of identifiers to the reference set. For UniProt, we corrected for deprecated or secondary Acs by translating to primary gene symbol and then to primary UniProt AC. We applied corrections to handle non-standard notations (e.g., extra dashes, Greek letters). We also accounted for the ambiguity in the mapping, i.e., if one foreign identifier may correspond to multiple reference identifiers we keep all target identifiers in OmniPath.

### Translation by homology to rodent species

The homology translation in pypath uses the NCBI HomoloGene database (NCBI Resource Coordinators, 2018). Because HomoloGene uses RefSeq IDs, the translation takes three steps: from UniProt to RefSeq, then to the homologous RefSeq and finally from RefSeq to UniProt. The success rate of this translation is around 80% for mouse and roughly 30% for rat (Dataset EV9). We translated the network data and the enzyme–PTM interactions from human to mouse and rat, the two most popular mammalian model organisms. In addition, we looked up PTMs in PhosphoSite (Hornbeck et al, 2015) which provides homology data for PTM sites. Then, we checked the residues in the UniProt sequences (UniProt Consortium, 2019) and discarded the ones that did not match. The homology-translated data are included also in the OmniPath web service.

### Data download and caching

At the database build, we download all input data directly from the original sources (Dataset EV1). Certain databases either temporarily or ultimately went offline; we deposited their data in the OmniPath Rescued Data Repository (https://rescued.omnipathdb.org/). Pypath contains the URLs for all resources used including the identifier translation tables. It automatically downloads, extracts, and preprocesses the data for each operation. Then, it stores the downloaded

data in a local cache directory which belongs to the user account on the computer. Once cache is created, *pypath* reads from it and performs the download only if requested by the user.

## Joint analysis of intra- and intercellular processes in *SARS-CoV-2* infection

The *NicheNet* method (Browaeys *et al,* 2019) was built, trained, and applied to a case study using interactions and annotations from *OmniPath* resources. This information was downloaded via our R package, *OmnipathR*.

### Network construction

NicheNet generates a model based on prior knowledge to describe potential regulatory effects of ligands on target genes. To reproduce their procedure, we first built three networks accounting for protein interactions of different categories retrieved from *OmniPath*:

1   Ligand–receptor network: We downloaded the "ligrecextra" network which specifically contains known interactions between ligands and receptors. In addition, we selected proteins annotated as *ligands* or *receptors* as their main "intercellular signaling role". Then, we extended this network with PPI whose source is a ligand and its target a receptor.
2   Signaling network: we retrieved PPI from the original *OmniPath* network (Türei *et al,* 2016), the "kinaseextra" network and the "pathwayextra" network.
3   Gene regulatory network: We selected the most reliable TF–target interactions from the *DoRothEA* dataset (confidence levels A, B, and C) and the literature curated "tf_target" dataset of the "transcriptional" network of *OmniPath* to be in line with the curation level of the ligand–receptor and signaling networks.

Then, we computed ligand–target regulatory potential scores based on the topology of our aforementioned networks, following the protocols described in the *NicheNet* original study and using its associated *nichenetr* package (Browaeys *et al,* 2019). Briefly, *Personalized PageRank* was applied on the union of the ligand–receptor and signaling networks considering every individual ligand as starting node. To estimate the impact of every ligand in the expression of target genes, a matrix containing the *PageRank* scores is multiplied by the weighted adjacency matrix of the gene regulatory network.

### Analysis of altered ligands and pathways

We applied our *OmniPath*-based version of *NicheNet* analysis on RNA-Seq data of a human lung cell line, *Calu3* (GSE147507) (Blanco-Melo *et al,* 2020). In this study, differential expression analysis at the gene level between controls and *SARS-CoV-2*-infected cells was carried out using the *DESeq2* package(Love *et al,* 2014). We selected over-expressed ligands (adjusted *P*-value < 0.1 and Log2 fold-change > 1) after *SARS-CoV-2* infection for further analysis. Then, we executed *Gene Set Enrichment Analysis (GSEA)* taking the Wald statistic and the hallmark gene sets from *MSigDB* (Liberzon *et al,* 2011) as inputs using the *fgsea* package (preprint: Korotkevich *et al,* 2016). Inflammatory response appeared as one of the top enriched sets. We therefore selected the leading edge genes of inflammatory response, i.e., genes contributing the most to the enrichment of this particular set, as

potential targets of the over-expressed ligands. We chose the inflammatory response genes, similarly to the original *NicheNet* study investigating the epithelial–mesenchymal transition-related genes (Browaeys *et al,* 2019), because these processes are likely to be regulated by extrinsic signals.

Ligand activity analysis on the aforementioned samples was conducted using the *nichenetr* package (Browaeys *et al,* 2019). We then selected the shortest paths between receptors (the ones interacting with the top predicted ligands) and transcription factors (the ones regulating the expression of the inflammatory target genes). These paths were exported to *Cytoscape* (Shannon *et al,* 2003) to generate Fig 5A.

## Intercellular communication in ulcerative colitis

### Intercellular interactions from OmniPath

We downloaded intercellular interactions using the *"import_intercell_network()"* method in *OmnipathR* and filtered for direct cell–cell connections: We discarded extracellular matrix proteins, extracellular matrix regulators, ligand regulators, receptor regulators, and matrix adhesion regulators and kept only membrane-bound (transmembrane or peripheral site of the membrane) proteins on the receiver side. This resulted in connections involving ligands, receptors, junction, adhesion, ion channel, transporter, and cell surface or secreted enzyme proteins.

### Single-cell RNA-Seq data processing

We downloaded the raw scRNA-Seq data and processed it according to Smillie et al (Smillie *et al,* 2019). 51 cell types have been characterized by average gene expressions in healthy ($n = 12$) state and non-inflamed UC ($n = 18$). A gene was considered expressed if its log2 expression value was above the mean minus 2 standard deviations of the expressed genes within the cell type.

### Specific interactions between cell types

We examined all possible connections among the selected 5 cell types. We considered interactions condition specific if they appeared either only in healthy or in UC, i.e., at least one member was expressed only in the given condition. We counted the unique PPIs between each cell pair in the two conditions separately (Fig 5B). We visualized the condition-specific connections from myofibroblasts to T cells on circos plots using the circlize R package (Gu *et al,* 2014). On these figures, we grouped similar ligands (e.g., CCR2 and CCR5) and merged the connections within groups. Then, we grouped the receptors by pathways defined in *SignaLink* (Fazekas *et al,* 2013) to improve biological insight and visual clarity (Fig 5C).

### Cell type-specific network of regulatory T cell and downstream pathway analysis

To highlight the downstream effect connections from myofibroblasts to regulatory T cells, we created a cell-specific signaling network and we carried out a pathway enrichment analysis. We used the *OmniPath Cytoscape* application (Ceccarelli *et al,* 2019) to combine the gene expression data with the *OmniPath* network. We limited the network to genes expressed in regulatory T cells. We selected the receptors targeted by condition-specific ligand–receptor connections in regulatory T cells. Finally, we pruned the network to the two steps neighborhood of the T cell-specific receptors. We

performed a pathway enrichment analysis on the network described above, using the online interface of the *Reactome* database with its default settings (hypergeometric test, Benjamini–Hochberg FDR correction, the human genome as the universe).

### Comparing OmniPath to other resources for cell–cell interaction analysis

For the protein interaction network, we downloaded *Pathway Commons* (Cerami *et al*, 2011), which is an integrated resource containing undirected protein–protein connections from public pathways and interaction databases. *Pathway Commons* was downloaded from the version https://www.pathwaycommons.org/archives/PC2/v12/PathwayCommons12.All.hgnc.sif.gz. Ligand, receptor annotations were derived from *Ramilowski et al* and were downloaded from https://fantom.gsc.riken.jp/5/suppl/Ramilowski_et_al_2015/data/PairsLigRec.txt. We run our pipeline for three different network–annotation combinations: (i) *Pathway Commons* network with *Ramilowski* annotations; (ii) *Pathway Commons* network with *OmniPath* ligand, receptor annotations; and (iii) *OmniPath* network with *OmniPath* ligand, receptor annotations.

## Data availability

*OmniPath* is available via the Python package *pypath* (https://github.com/saezlab/pypath), the web resource (https://omnipathdb.org), the R/Bioconductor package *OmnipathR* (https://saezlab.github.io/OmnipathR), the *omnipath* Python client (https://github.com/saezlab/omnipath), and the *OmniPath* Cytoscape plug-in (Ceccarelli *et al*, 2019). In addition, *pypath* is able to export the network and the enzyme–PTM databases in *BEL (Biological Expression Language)* format (Hoyt *et al*, 2018b), as well as to generate input files for *CellPhoneDB*. The BEL format databases are available in *BEL Commons* (Hoyt *et al*, 2018a). Code is licensed open source (GPLv3 or MIT). *Pypath* builds the *OmniPath* databases directly from the original resources, hence it gives the highest flexibility for customization and the richest API for queries and manipulation among all access options. Furthermore, it is possible to convert each database to a plain data frame and export in a tabular format. *Pypath* also generates the web resource's contents which is accessible for any HTTP client at https://omnipathdb.org. Information about the resources is available at https://omnipathdb.org/info. *OmnipathR* and the *OmniPath Cytoscape* plug-in work from the web resource data with convenient postprocessing features. All data in *OmniPath* carry the licenses of the original resources (Dataset EV12), for profit users can easily limit their queries to fit the legal requirements. We maintain a directory of workflows and tutorials at https://workflows.omnipathdb.org/.

Apart from the figures presented in this paper, further regularly updated statistics and visualizations are available at https://insights.omnipathdb.org.

A Python and R package for producing the figures and tables of this paper is available at https://github.com/saezlab/omnipath_analysis. The code to build and train the *NicheNet* method (Browaeys *et al*, 2019) exclusively using *OmniPath* resources as well as to reproduce the *SARS-CoV-2* case study is freely available at https://github.com/saezlab/NicheNet_Omnipath. The code for building cell type-specific inter- and intracellular networks is available at https://github.com/korcsmarosgroup/uc_intercell.

Expanded View for this article is available online.

## Acknowledgements

This work was partially supported by the JRC-COMBINE, partially funded by Bayer, the European Union Innovative Medicines Initiative TransQST (agreement No. 116030), the Federal Ministry of Education and Research (BMBF, Computational Life Sciences grant no. 031L0181B), the DFG (Deutsche Forschungsgemeinschaft / German Research Council; Funding code: SA 3554/1-2), to J.S.R. T.K. and D.M. were supported by the Earlham Institute (Norwich, UK) in partnership with the Quadram Institute (Norwich, UK) and strategically supported by the UKRI Biotechnological and Biosciences Research Council (BBSRC) UK grants (BB/J004529/1, BB/P016774/1, and BB/CSP17270/1) and by a BBSRC ISP grant for Gut Microbes and Health BB/R012490/1 and its constituent projects, BBS/E/F/000PR10353 and BBS/E/F/000PR10355. L.G. and M.O. were supported by the BBSRC Norwich Research Park Biosciences Doctoral Training Partnership grant number BB/M011216/1. Thanks to John P Thomas, Robin Broeways, Yvan Saeys, Mirjana Efremova, Daniel Domingo-Fernandez, Charles Tapley Hoyt, Lu Li and Paul D Thomas for their helpful feedback and discussions. Open Access funding enabled and organized by Projekt DEAL.

## Author contributions

Design and development of *pypath* and *OmniPath* and creating descriptive figures and tables: DT; *OmnipathR* package: AV, DT, and AG; Designing and performing case study on *SARS-CoV-2* infection data: AV; Designing and performing case study on ulcerative colitis: LG and DM; Development of *pypath* and visualization of the database contents: NP-E, OI, and LG; *omnipath* Python module: MK; Supervision of *omnipath* Python module: FT; Tutorials and analysis of coverage of ligand–receptor databases on textbook data: MO; Supervision of the project: JS-R and TK; Manuscript writing: All authors.

## Conflict of interest

JSR receives funding from GSK and Sanofi and consultant fees from Travere Therapeutics.

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
