## [Review Process File · Molecular Systems Biology]

Integrated intra- and intercellular signaling knowledge for multicellular omics analysis

Dénes Türei, Alberto Valdeolivas, Lejla Gul, Nicolás Palacio-Escat, Michal Klein, Olga Ivanova, Marton Olbei, Attila Gábor, Fabian Theis, Dezso Modos, Tamas Korcsmaros, and Julio Saez-Rodriguez
DOI: [10.15252/msb.20209923](https://doi.org/10.15252/msb.20209923)

Corresponding author: Julio Saez-Rodriguez (julio.saez@bioquant.uni-heidelberg.de)

Review Timeline:

Submission Date:	14th Aug 20
Editorial Decision:	23rd Sep 20
Revision Received:	19th Dec 20
Editorial Decision:	27th Jan 21
Revision Received:	11th Feb 21
Accepted:	15th Feb 21

Editor: Maria Polychronidou

Transaction Report:

RE: MSB-20-9923, Integrated intra- and intercellular signaling knowledge for multicellular omics analysis

Thank you again for submitting your work to Molecular Systems Biology. We have now heard back from the three referees who agreed to evaluate your study. As you will see below, the reviewers acknowledge that the proposed workflow/resource seems potentially useful for the field. However, they raise a series of concerns, which we would ask you to address in a major revision.

Without repeating all the issues listed below, some of the more fundamental issues are the following:

- As reviewers #1 and #2 mention, a direct comparison to alternative resources and methodologies needs to be included. The advantages and improvements that OmniPath offers need to be better demonstrated.
- As the reviewers mention, in its current form the study does not provide sufficient biological insights. The case studies needs to be extended in order to provide concrete biological insights and conclusively show that OmniPath allows going beyond what is possible to achieve with alternative approaches. We think that such analyses would significantly enhance the impact of the work.
- Reviewer #3 recommends providing a clear workflow, in order to make sure that the resource can be easily adopted by future users.

Please let me know in case you would like to discuss in further detail any of the issues raised. All issues raised by the referees would need to be satisfactorily addressed.

Reviewer #1:

The paper by Turei et al describes an update of the Omnipath resource, a previously published integrated database, to better facilitate research on intercellular communication. This updated version of OmniPath contains now several resources covering inter- and intracellular signal transduction and gene regulatory interactions. Hereby, this resource can be used as a single access point for approaches that aim to model inter- and intracellular signaling processes (and ultimately multicellular systems). To this end, novel recent databases are added, as well as novel annotations are defined to describe in more detail all the constituents of intercellular communication.

This results in 5 "new" sub-databases of omnipath to be used for better interpreting intercellular communication analyses. In addition to this, the authors describe several other interesting features, such as the addition of rich and curated annotations, novel data query possibilities (e.g. a new R/Bioconductor package), and homology translation between human and mouse/rat. In the main paper and supplementary information, they also provide a detailed description of the several resources at the basis of OmniPath 2, and how these resources compare to each other. The corresponding software is documented and user-friendly. Computational biologists that want access to up-to-date resources will benefit from using this single access point and richly annotated resource.

The usefulness of the resources is subsequently illustrated on two case studies. The major claim made by the authors is that this is a useful, integrated database for multicellular omics analyses.

While it is true that Omnipath becomes more useful in this way to study intercellular communication, the novelty of the manuscript is limited, and does not contain much technical or biological novelty. It is rather a "resource" type of article, which describes an additional feature of an existing data source. In addition, it is not immediately clear how this resource improves upon the state-of-the-art, especially in the field of modelling intercellular communication as no direct comparisons to other methods are incorporated. For example, for the first case study, the authors replaced the standard NicheNet network by their Omnipath derived network, but one would expect them to at least compare to the standard NicheNet network. Also comparisons to using other integrated databases for this task (such as PathwayCommons) are lacking. PathwayCommons also provides both signaling and gene regulatory interactions and could thus in theory also be used as single access point for modeling multicellular systems. In the Discussion section, the authors already briefly discuss some differences (in representation: activity flow versus process description), but I think it would be interesting to discuss the following questions: 1) why someone would benefit more from using OmniPath than using Pathway Commons for modeling multicellular systems? 2) are there different resources used at the basis of the integrated databases?

One particular aspect that was unclear from the manuscript was to which extent cell type or tissue specificity is taken into account. If users are interested in specific cell types or tissues, can they then easily filter this information from Omnipath and obtain more specific information? It is my impression that the current case studies combine cell-type specific gene expression information with cell-type agnostic Omnipath derived relationships. If these could be made more specific, it would be a way to remove false positive interactions and obtain more tissue- or cell type specific interactions.

While the two use cases at the end of the paper describe potential biological applications, I find the resulting findings rather vague, and I guess biologists would not learn too many new things from it.

For case study one, a lot of questions remain:

- How do the results of this NicheNet analysis with the 'OmniPath model' compare to the default NicheNet model? Are different ligands and target genes predicted? Are ligand activity values and signaling links different? I am not asking for a systematic quantitative comparison because the ground truth is unknown, but I think it could be of interest to readers to compare both outputs (in the corresponding Supplementary Note). Even a comparison to a 'PathwayCommons model' might be interesting.
- Could the authors motivate why they choose to run the NicheNet ligand activity analyses on the leading edge genes of the 'inflammatory response' set instead of on all differentially expressed genes? And also why 'level D' interactions from DoRothEA and the 'tf_target' network were not used as input for the gene regulator network?

For the second case study it seems not surprising that the JAK-STAT pathway pops up, as this is very well studied in the literature, but this is nothing new for a biological audience, so I would appreciate some more specific and maybe novel results, ideally also taking into account cell type specificity.

Minor remarks concerning presentation and style:

- o r140-143 and r294-299 felt "out of place".
- o the differences between this updated version of the OmniPath database and the first release were not directly clear to me.
- o Also the distinction between pypath and OmniPath was not directly clear to me.

Reviewer #2:

The authors present a new meta-database that has collected all information available in other databases regarding the molecular details of inter- and intra-cell signalling and includes in-depth annotations. They present two case studies in prioritising receptor-ligand interactions relevant to coronavirus infection and demonstrating changes in cell-cell networks in ulcerative colitis. Overall the manuscript is clear and well-written, and this resource will be valuable for the community as it provides a one-stop-shop for all this information. The previous version of Omnipath is also widely used by the community and this upgrade will serve to further increase its impact for use also by the single-cell and spatial transcriptomics community as well.

Major points

I am not entirely convinced that the case studies showcase the value of the resource as they currently are described. Specifically, to demonstrate its, it would be good to see how using Omnipath compares to the next biggest resource for this kind of data. Or for the resulting networks used to make the conclusions it would be good to show the number of resources from which the nodes/data points were extracted, to make the case that this integrated resource is necessary for this analysis.

In addition, the enrichment analyses performed in the coronavirus story should use as a background the 117 differentially expressed receptors rather than the whole backgrounds set, to convince that the 12 genes prioritised through use of Omnipath and NicheNet have indeed identified the ones among the 117 that were the most relevant, i.e. with enrichment in inflammatory processes, JAK/STAT pathway etc.

Minor points

It is not clear if this is included amongst the annotations already, but it would be good to have a very clear indication of the source of data, i.e. computationally inferred, manually annotated, complexes etc so that the user can filter with respect to quality. e.g. CellPhoneDB is much more accurate than the other resources, as it was manually curated and so provides better insight since it doesn't cloud the cell-cell networks with as much noise.

It would be nice to add some more information with respect to the definition of the complexes. At the moment all it says is "A complex is defined by its unique combination of members". Are all pull-downs e.g. considered complexes? Are complex components supposed to be universally present across multiple pull-down experiments? Some more information would be useful here.

I found the legends boxes in figures 2a,2b,2d and 3d confusing.

Reviewer #3:

Integrated intra- and intercellular signaling knowledge for multicellular omics analysis
Dénes Túrei, Alberto Valdeolivas, Lejla Gul, Nicolàs Palacio-Escat, Olga Ivanova, Attila Gábor, Dezső Módos, Tamás Korcsmáros, Julio Saez-Rodriguez*
MSB (Sept, 2020)

Summary

The authors continue to build on their previous work, developing OmniPath as a flexible and useful tool to capture molecular interactions from a wide-range of publicly available databases (Túrei et. al 2016). In this manuscript, OmniPath has been updated to incorporate integration of networks of inter- and intracellular signaling molecules. OmniPath was used to generate molecular inputs for cell-cell communication modeling for the two case studies, in concert with other analytical software, by combining information from numerous databases. They demonstrate its use with two case studies: 1) bulk RNA-seq data used to identify networks of differentially upregulated inflammatory ligands and their target genes of a Calu-3 lung epithelium cell line infected with SARS-Covid-2. They found proteins associated with the JAK-STAT and MAPK pathway pathways, which is supported by literature. 2) They used inter- and intracellular molecular interaction networks from single cell RNA-sequencing of 5 intestinal niche cells for ulcerative colitis, defining inflammatory signals underlying the diseased state. They suggested that the inflammatory signaling in ulcerative colitis derives from Treg and myofibroblast cell interactions, mediated by TLR pathways.

General remarks

The manuscript serves as an update to the capabilities of the OmniPath software, originally presented in Túrei et. al 2016. The case studies rely heavily on existing literature for source data and validation for the conclusions drawn from their workflows that depend on their software. It is written for an audience that is familiar with cell-cell communication network modeling. The authors do an excellent job of placing their software in the ecosystem of available molecular interaction databases, quantifying the coverage of each of their databases in comparison to others. They show a clear advantage to using their unifying tool. The case studies provide the reader with examples of how OmniPath could fit into an analysis

workflow, as well as that value of combining inter- and intracellular signaling interactions.

Major Points

Overall, the main criticism is that the format of the manuscript does not clearly lay out use of the software, but instead emphasizes its place in the field's ecosystem of tools, which is important, but in the opinion of this reviewer, a secondary objective of a tool-based manuscript. A focus on a clear workflow with more detail for the case studies would make the software more accessible and highlight its importance in the field.

In general, the case studies serve as a demonstration of the software, but rely heavily on the results and interpretations of other published literature. To increase the impact of this paper, more work could be done to develop novel mechanisms using OmniPath. However, the emphasis of this manuscript is on tool development, so this is a secondary concern.

Suggestions for improvement:

- Well-written introduction. Can the authors include some exceptional example publications that use OmniPath, which can help expand the reader's appreciation for use-cases?
- In the results section, briefly outline OmniPath's place in the tool ecosystem, but move major comparisons of OmniPath to other tools to supplementary (or last part of the results?).
- Put greater emphasis on each of the case studies (in light of the updates to the software - inter/intracellular signaling integration) and expand the methodology used in each with sufficient detail to guide the reader on its use and its integration with other software tools, from the perspective of a workflow (figures can illustrate and summarize this).

E.g. bring more of SNote 1 into the main text, develop a SNote 2 to support Case Study 2 as in 1, and bring some or all of SFig 6 into the main text. Also a diagram of the workflow of the case studies would be essential, this can serve as a template for future studies.

E.g. L273: Please go into greater detail describing the evidence for myofibroblast regulation of Treg pro-inflammatory responses to UC, highlighting key molecules and referencing STable 11.

- support the selection of candidate molecules (and other decisions) by reporting statistics
- E.g. L248: How did the authors threshold their analysis for 12 ligands? Could they provide a justification? Were there any interesting down-regulated hits? Same for L273 candidates.

L225: As a demonstration and validation of OmniPath, these cases are useful and they agree with published literature. For both case studies, the inflammatory response gene ontology category is a very broad and expected one. Were there other gene sets that were less enriched but more descriptive or specific to the experimental condition indicating novel findings?

Although the analysis was built from bulk data, is it possible for the authors to propose a map of cell-cell interactions based on the receptors and ligands involved in the described inflammatory response? For example, for the candidate chemokines, can they be mapped to putative leukocyte cell populations? e.g. IL23A to macrophages or dendritic cells, etc.

L256: The strategy and workflow in Case 2 are much clearer than Case 1. Perhaps a supplementary note could add some details (or add to the main text) about how the cell-cell interaction networks and disease-specific changes were modeled. As in SNote 1 for Case 1, the implications of these findings could be better explored as well.

Minor Points

L36: In the final paragraph of the introduction, it would be helpful to describe briefly the biological rationale behind the choices of the two case studies.

L56: The authors mean 'some key players' instead of 'key players'?

L144+: Can the authors introduce each database type by placing the value of its information into context of the field to better guide the reader in building their own workflows with the software?

L182: Fig 3b and 3c. Perhaps there is a better way to represent this information? It is difficult to distinguish categories at a glance with so many different colors. Perhaps with identifiable symbols? Are these colors appropriate for forms of color-blindness?

L196: For clarification, does the OmniPath protein complexes database contain information on states of the complex based on present constituent members, or be used to generate such information?

L228: Figure 4a is meant to summarize ways in which OmniPath can integrate with other software tools, but is unclear and difficult to interpret, particularly due to the layout of small arrows. This is in contrast to 4c, where the workflow is graphically clearer. As 4a is an important summary graphic, recommend redesign.

L251: Authors state in the text that Inflammatory ligands have downstream JAK-STAT targets - please report these targets, in the text and/or in Fig 4.

L253: The authors propose the JAK-STAT pathway could provide drug targets for Covid-19 treatment - can they expand on this, perhaps by leveraging their network-based approach, and propose candidate drug targets? Is it possible to rank drug targets, say based on limited off-target genes, or by connecting to DGldb, etc. In SNote 1, Ruxolitinib was mentioned as a potential JAK-STAT-interacting drug, but this was already introduced by the authors of the source data (Blanco-Melo et al.).

L254: The authors reported true-positive results about inflammatory pathways. Would they be able to argue if there are other important pathways reported to be promoted by SARS-CoV-2 infection that has not been captured here? They can argue the reason in the discussion section.

L268: Missing the word "cells" after "Treg".

L273: "TLR431 and TLR3 pathways³² were upregulated in UC." In Treg cells?

L288+ (Discussion section): Can the authors suggest best practices that would improve the addition of new resources to future iterations of OmniPath?

How can database-derived interaction networks such as those built with OmniPath start to incorporate magnitude of cell signaling events?

Consider suggesting how to extend case study results by experimentation, or other opportunities to build on this work. Spatial annotation suggested by the authors is an excellent example. Can the authors expand further?

For further discussion, what major challenges remain, either for the authors, the researchers generating the resources, or the end-users?

L622: The Blanco-Melo, D. 2020 Microbiology reference from the main text (Reference 23) SNote 1 (Reference 1) seems to be incorrect. It was originally published under that title in BioRxiv, but later peer-reviewed and published under a new title in Cell here: doi: 10.1016/j.cell.2020.04.026

L730: How do you interpret and threshold NicheNet's ligand-target regulatory potential and ligand-receptor interaction potential statistics in SFig6c,d? How do these statistics compare to ligand-target/ligand-receptor statistics for those not chosen? Clarification of these in the text would strengthen the interpretation of the analysis for the reader.

Can the authors report statistics on conflicting annotations from different databases as part of their comparison to other tools and molecular coverage, perhaps in the supplementary material?

Thank you for your feedback and sharing the comments of the reviewers of our manuscript “*Integrated intra- and intercellular signaling knowledge for multicellular omics analysis*”, and by the opportunity to submit a revised version. We believe the reviewers’ suggestions helped us to improve our manuscript significantly, and this revision is addressing all of the points they raised, including the ones you highlighted in your letter:

As reviewers #1 and #2 mention, a direct comparison to alternative resources and methodologies needs to be included. The advantages and improvements that OmniPath offers need to be better demonstrated.

To better show the value of OmniPath compared to other resources, we added a new table (Table 1) and ran the case studies with alternative resources. Of note, no other single resource can provide all elements of knowledge necessary for these analyses, except OmniPath. In addition, we rewrote the Discussion to further highlight the novelties of OmniPath.

As the reviewers mention, in its current form the study does not provide sufficient biological insights. The case studies needs to be extended in order to provide concrete biological insights and conclusively show that OmniPath allows going beyond what is possible to achieve with alternative approaches. We think that such analyses would significantly enhance the impact of the work.

We also largely extended on the biological context of the case studies. Most of this is within the Extended View to keep the focus of the paper in the tool and resources.

Reviewer #3 recommends providing a clear workflow, in order to make sure that the resource can be easily adopted by future users.

We have now created a collection of workflows, that are available at <https://workflows.omnipathdb.org/>. Nine of these workflows are accompanied by didactic tutorials, facilitating the access and usage of data analysis with OmniPath. This was led by Marton Olbei, who is subsequently a coauthor of the paper.

In addition, after discussions with the authors of NicheNet, Robin Browaeys and Yvan Saeys, we created a full integration of NicheNet and OmniPath in our R package. This not only gives much more convenience and flexibility for the users, and also addresses the point from Reviewer #3 about the need for more accessible workflows.

We would like to mention a few more new aspects:

OmniPath is under continuous development, and we always keep including new resources into OmniPath. As an example, while working on the revision we have added four freshly published

resources for cell-cell communication: CellTalkDB, CellChatDB, connectomeDB2020 and Wojtowicz et al. 2020.

In addition, OmniPath is used by many computational and modelling groups. Michal Klein and Fabian Theis spontaneously contributed with a new Python web service client, as they leverage OmniPath in their own research on machine learning and single-cell data. We have included this client, that links OmniPath to the scanpy framework, in the manuscript, with a tutorial under workflows.omnipathdb.org, and they are now coauthors of the paper.

As a testament to the value of OmniPath for the community, it recently became a part of the COVID DiseaseMap, a large-scale community effort to integrate database knowledge about the COVID-19 disease mechanisms (<https://www.biorxiv.org/content/10.1101/2020.10.26.356014v1>).

We attach a point by point response to the reviewers' comments.

We hope this revised version will be acceptable for publication in Molecular Systems Biology.

Response to reviewers

We thank the reviewers for the thorough assessment of our manuscript and the many helpful comments which directed us towards improvements. Please find below our answers to the comments and questions with references to the revised version of the manuscript.

Reviewer 1

The paper by Turei et al describes an update of the Omnipath resource, a previously published integrated database, to better facilitate research on intercellular communication. This updated version of OmniPath contains now several resources covering inter- and intracellular signal transduction and gene regulatory interactions. Hereby, this resource can be used as a single access point for approaches that aim to model inter- and intracellular signaling processes (and ultimately multicellular systems). To this end, novel recent databases are added, as well as novel annotations are defined to describe in more detail all the constituents of intercellular communication.

This results in 5 "new" sub-databases of Omnipath to be used for better interpreting intercellular communication analyses. In addition to this, the authors describe several other interesting features, such as the addition of rich and curated annotations, novel data query possibilities (e.g. a new R/Bioconductor package), and homology translation between human and mouse/rat. In the main paper and supplementary information, they also provide a detailed description of the several resources at the basis of OmniPath 2, and how these resources compare to each other. The corresponding software is documented and user-friendly. Computational biologists that want access to up-to-date resources will benefit from using this single access point and richly annotated resource.

The usefulness of the resources is subsequently illustrated on two case studies. The major claim made by the authors is that this is a useful, integrated database for multicellular omics analyses.

While it is true that Omnipath becomes more useful in this way to study intercellular communication, the novelty of the manuscript is limited, and does not contain much technical or biological novelty. It is rather a "resource" type of article, which describes an additional feature of an existing data source.

1. We are happy to read that the reviewer agrees with us that the resources provided in this manuscript are useful. And we indeed see this as a resource. Before OmniPath, no meta-resource for intercellular communication was available.

We understand that at first sight the novelty might seem limited, as the design of OmniPath might seem straightforward. However, we believe in a number of important points that bring significant novelty to the field. We rewrote a large part of the Discussion to emphasize these more. In particular,

"By defining the transmitter, receiver and mediator roles, we layed out a new conceptual framework to describe intercellular communication and generalized the terms of ligand and

receptor (Dataset EV10). This framework allows *OmniPath* to combine diverse resources in a uniform way. In *OmniPath* the intercellular annotations and the network connections are independent from each other, achieving great flexibility in their combination.”

This flexibility can be extended to other network resources, for example one can use PathwayCommons with *OmniPath*'s annotations to create intercellular communication networks.

In our data integration work we put this framework into practice and we aimed for the broadest integration of this type of knowledge in the field. In particular, we combine more resources about intercellular communication than any other resource we are aware of. With *OmniPath*, researchers studying intercellular communication are able to easily select and use the combination of resources which works the best for their purposes. Furthermore, we manually revised tens of thousands of annotations, thereby improving the quality. For these reasons we believe that this manuscript does provide conceptual and technical advances as a meta-resource for systems biology.

In addition, it is not immediately clear how this resource improves upon the state-of-the-art, especially in the field of modelling intercellular communication as no direct comparisons to other methods are incorporated. For example, for the first case study, the authors replaced the standard NicheNet network by their *OmniPath* derived network, but one would expect them to at least compare to the standard NicheNet network.

2. We thank the reviewer for noting that this was not clear, and we have clarified this in the revised version.

NicheNet is a tool, and not a resource: the authors manually assembled a network for the paper, and they have no system to update and distribute its prior knowledge corpus. Indeed, *OmniPath* complements and supports NicheNet and other tools: The authors of NicheNet needed to put a great effort in collecting and organizing the required prior knowledge, despite this not being the focus of their work, and they could have spared this if the *OmniPath* intercell database existed that time. In fact, during the revision we talked to the lead developers of NicheNet (Robin Browaeys and Yvan Saeys), and we agreed that the software infrastructure of *OmniPath* can support NicheNet users with prior knowledge better, in terms of maintainability, flexibility and reproducibility.

We implemented in *OmniPathR* (the R/Bioconductor package) a number of methods to build NicheNet input databases not only from *OmniPath* but also directly from the original resources in a fully customizable way. This major development (2,000 lines of code) allows NicheNet users to have an updated network for their analyses by just calling a function in *OmniPathR*. With this update, we provide not only a close integration of *OmniPath* and NicheNet but also R interfaces to further 20 resources. We have added this information to the revised manuscript: *“The *OmniPathR* R client implements a full integration with NicheNet, a method for prioritizing ligands affecting cells based on transcriptomics data: a single *OmniPathR* function can be used to generate all inter- and intracellular knowledge required for NicheNet.”*

Also comparisons to using other integrated databases for this task (such as PathwayCommons) are lacking. PathwayCommons also provides both signaling and gene regulatory interactions and could thus in theory also be used as single access point for modeling multicellular systems. In the Discussion section, the authors already briefly discuss some differences (in representation: activity flow versus process description), but I think it would be interesting to discuss the following questions: 1) why someone would benefit more from using OmniPath than using Pathway Commons for modeling multicellular systems? 2) are there different resources used at the basis of the integrated databases?

3. We agree that the unique points of Omnipath compared to alternative resources should be discussed more thoroughly, and in particular clarify why one can not use other databases such as PathwayCommons for the analysis that can be done with OmniPath, and vice versa.

OmniPath and PathwayCommons are both integrative resources but they are different in i) the original resources they integrate, ii) how the process and represent the data and iii) their scope and purpose. We added more details about these differences in Appendix 1 (where we also moved some parts of the Discussion section). These are important differences which determine the usability of PathwayCommons and OmniPath in downstream methods.

As the Reviewer mentions, PathwayCommons provides signaling and gene regulatory interactions and these parts can serve as an alternative to OmniPath. However PathwayCommons doesn't annotate the intercellular communication roles of proteins, hence this data must come either from OmniPath (or the databases inside OmniPath; listed in Figure 2d) before it can be used for analyses such as those described here.

We also ran the IBD case study using network data from PathwayCommons to show the effect of the different database knowledge on downstream analysis (see the "*Comparing Omnipath to other resources for cell-cell interaction analysis*" subsection in Results and Appendix Figure S3 for details).

One particular aspect that was unclear from the manuscript was to which extent cell type or tissue specificity is taken into account. If users are interested in specific cell types or tissues, can they then easily filter this information from Omnipath and obtain more specific information? It is my impression that the current case studies combine cell-type specific gene expression information with cell-type agnostic Omnipath derived relationships. If these could be made more specific, it would be a way to remove false positive interactions and obtain more tissue- or cell type specific interactions.

4. Interaction database data is almost always cell type agnostic. As we mention in the manuscript, prior knowledge interaction networks are usually contextualized using omics datasets as, for instance, described in our case studies .

It is also possible to build cell type or tissue specific networks using generic expression data, for example from Human Protein Atlas (HPA). HPA is available in the OmniPath Annotations

database and, with our Bioconductor package, it can be easily combined with the interaction network. We added this information to the Discussion and provide an example for this in one of our newly created tutorials (*“Close connection to the analysis of omics data”* subsection in Results).

While the two use cases at the end of the paper describe potential biological applications, I find the resulting findings rather vague, and I guess biologists would not learn too many new things from it.

5. We thank the Reviewer for pointing out that the use case data interpretation sections were too brief. Accordingly, we have extended the biological interpretation of the case studies. We also note that the interpretation and discussion of the result of these case studies are consciously limited as we think going further in this direction would be out of the scope of this paper. The purpose of the case studies is to illustrate how OmniPath can be useful.

For case study one, a lot of questions remain

- How do the results of this NicheNet analysis with the 'OmniPath model' compare to the default NicheNet model? Are different ligands and target genes predicted? Are ligand activity values and signaling links different? I am not asking for a systematic quantitative comparison because the ground truth is unknown, but I think it could be of interest to readers to compare both outputs (in the corresponding Supplementary Note). Even a comparison to a 'PathwayCommons model' might be interesting.

6. As mentioned above (Answer 2), we now provide a function to directly build prior knowledge networks from the resources used in the NicheNet paper and from OmniPath. These networks are directly retrieved in the input format used in the NicheNet workflow. Hence, an external user can now run the NicheNet method with any custom combination of these resources as described in the new workflows (*“Close connection to the analysis of omics data”* subsection of Results) including the combination used in the original NicheNet paper.

NicheNet results highly rely on the selected prior knowledge network. The OmniPath network and the original NicheNet network are different in size and scope (See OmniPath-NicheNet workflow: <https://workflows.omnipathdb.org/nichenet1.html>). Thus, a comparison between results would not be very informative and could only be conducted in a qualitative way in absence of a suitable ground truth, as pointed out by the reviewer.

The PathwayCommons data can not be used directly for NicheNet because it lacks the intercellular annotations (see Answer 2 and the *“Comparing OmniPath to other resources for cell-cell interaction analysis”* subsection in Results). We have instead compared our OmniPath-based network to PathwayCommons in the second case study (*“Alteration of intercellular communication in ulcerative colitis”* subsection in Results).

- Could the authors motivate why they choose to run the NicheNet ligand activity analyses on the leading edge genes of the 'inflammatory response' set instead of on all differentially expressed genes?

7. The goal of this case study is to study cell-cell communication through the integration of inter- and intracellular interactions. In particular, the potential effect of ligands in the regulation of the expression of a given set of target genes after SARS-CoV-2 infection.

We believe that genes involved in the inflammatory response are appropriate in our case study for two main reasons: 1) the inflammatory response involves the coordinated communication of different cells through cascades of molecular signals and 2) an exacerbated inflammatory response is observed in COVID-19 most severe cases. We therefore consider that the expression of genes involved in the inflammatory response is possibly affected due to communication with neighboring cells. In our opinion, this assumption cannot be extended to all differentially expressed genes, since their expression can change due to the intra-cellular action of the virus and its proteins. Such a targeted analysis is the natural way to use NicheNet: in their original publication, the NicheNet authors selected a set of genes involved in a partial epithelial to mesenchymal transition program as potentially regulated by cell communication.

We clarify this in the Methods section where we added:

We chose the inflammatory response genes, similarly to the original NicheNet study investigating the epithelial-mesenchymal transition related genes⁸, because these processes are likely to be regulated by extrinsic signals.

In addition, we created a new discussion section in the Appendix 2 to elaborate on our choices and argue about the potential limitations of this case study.

We further clarify the limitations of inflammatory response in Answers 25 and 36.

And also why 'level D' interactions from DoRothEA and the 'tf_target' network were not used as input for the gene regulatory network?

8. We decided to select the more reliable TF-target interactions to be in line with the other networks used in the case study (ligand-receptor and signaling), which contain high-confidence interaction. DoRothEA's level D and E contain lower confidence TF-target interactions as described in (Garcia-Alonso et al *Genome Research* 2019). OmniPath's 'tf_target' network directly integrates literature curated resources, most of them are also part of DoRothEA (Datasets EV7 and EV8). We clarify this in the Methods section: *“we selected the most reliable TF-target interactions from the DoRothEA dataset (confidence levels A, B and C) and the literature curated “tf_target” dataset of the “transcriptional” network of OmniPath to be in line with the curation level of the ligand-receptor and signaling networks.”*

For the second case study it seems not surprising that the JAK-STAT pathway pops up, as this is very well studied in the literature, but this is nothing new for a biological audience, so I would appreciate some more specific and maybe novel results, ideally also taking into account cell type specificity.

9. Indeed, the JAK-STAT pathway is well established, and in fact we see this as a validating result. We agree with the Reviewer and, in the revised version, we highlighted the novel results we got in this use case and added more details to the Results. Our work provided a detailed and novel cell-cell interaction network with cell-specific downstream analysis. As we compared these networks between healthy and diseased conditions, we pointed out the mechanisms (ie, specific ligand-receptor connections and downstream pathways) that significantly changed between for example myofibroblasts and Treg cells. This new molecular insight explains how the previously observed anti-inflammatory role of Treg cells is rewired in ulcerative colitis due to changes initiated by myofibroblasts. Emphasising the novelty of the use case, we benchmarked OmniPath with other source datasets frequently used for similar studies. This benchmark clearly demonstrates now the increased coverage with decreased false positive interactions for such analysis.

Minor remarks concerning presentation and style

o r140-143 and r294-299 felt "out of place".

10. We agree that they were not well connected; we have now largely rewritten the Discussion, and moved some of this text to the Appendix 1, and we think that it is now better connected.

o the differences between this updated version of the OmniPath database and the first release were not directly clear to me.

11. Indeed this might not be clear enough, and we have added a sentence in the 3rd paragraph of the introduction where we first mention the original OmniPath: "The first version of OmniPath focused on literature curated intracellular signaling pathways". Furthermore, we modified the first paragraph of the discussion: *In the first version of OmniPath⁹ we built a comprehensive knowledge of intracellular signaling pathways with the aim of providing prior knowledge for modeling methods. Here, we present a major redesign of this resource, offering a single-access point to over 100 resources containing prior knowledge of not only intra- but also intercellular processes.*

o Also the distinction between pypath and OmniPath was not directly clear to me.

12. We apologize if this was not clear. We modified Figure 1b to visually separate pypath and OmniPath, added explicit statements to the caption and text around Figure 1, we hope this makes it more clear.

Reviewer 2

The authors present a new meta-database that has collected all information available in other databases regarding the molecular details of inter- and intra-cell signalling and includes in-depth annotations. They present two case studies in prioritising receptor-ligand interactions relevant to

coronavirus infection and demonstrating changes in cell-cell networks in ulcerative colitis. Overall the manuscript is clear and well-written, and this resource will be valuable for the community as it provides a one-stop-shop for all this information. The previous version of Omnipath is also widely used by the community and this upgrade will serve to further increase its impact for use also by the single-cell and spatial transcriptomics community as well.

We thank the reviewer for his positive words and we are glad that the reviewer finds the work useful.

Major points

I am not entirely convinced that the case studies showcase the value of the resource as they currently are described. Specifically, to demonstrate its, it would be good to see how using Omnipath compares to the next biggest resource for this kind of data.

13. We appreciate that this was not clear in our manuscript, as this was also raised by Reviewer 1. So far no resource offers both the intercellular communication annotations and the intracellular pathways and transcriptional regulation knowledge. Hence, there is no "next biggest resource for this kind of data" and such a benchmark is not possible. We have clarified this point in the text, and also included a table (Table 1) comparing the existing resources (see Answer 6 to Reviewer 1).

Or for the resulting networks used to make the conclusions it would be good to show the number of resources from which the nodes/data points were extracted, to make the case that this integrated resource is necessary for this analysis.

14. We thank the Reviewer for this idea, to directly show how the combination of resources results in a more complete knowledge. We created a new table (Dataset EV13) where we list all interactions in the first case study with the original resources for each interaction.

In addition, the enrichment analyses performed in the coronavirus story should use as a background the 117 differentially expressed receptors rather than the whole backgrounds set, to convince that the 12 genes prioritised through use of Omnipath and NicheNet have indeed identified the ones among the 117 that were the most relevant, i.e. with enrichment in inflammatory processes, JAK/STAT pathway etc.

15. We thank the reviewer for this suggestion. We carried out the enrichment analyses as proposed, i.e. we conducted a hypergeometric test on our 12 prioritised ligands using as a background the 117 over-expressed ligands after SARS-CoV-2 infection of the Calu-3 cell line. As prior knowledge, we used the list of curated pathways from MsigDB. The pathways with a p-value < 0.05 are shown below:

We found some pathways directly related to cytokines and therefore to our list of 12 prioritised ligands (IL23 Pathway, CXCR3 Pathway and IL10 Pathway). The NFKB pathway appears as the most enriched process. We already described how this pathway is connected to some of our predicted ligands and its activation during SARS-CoV-2 infection. Toll-like receptors are a class of proteins that are well known for their key role in the innate immune system. In addition, we retrieved an enrichment in the dilated cardiomyopathy pathway and in the HSP27 pathway. Interestingly, it has been postulated that the sustained immune activation upon SARS-CoV-2 infection increases the risk of developing dilated cardiomyopathy in COVID-19 patients (<https://pubmed.ncbi.nlm.nih.gov/32536978/>). Passive immunization using anti-HSP27 antibodies has been suggested as a potential treatment against the inflammatory complications of SARS-CoV-2 infection (<https://www.ncbi.nlm.nih.gov/pmc/articles/PMC7407440/>). We included these results in the Appendix 2 and Appendix Figure S2.

Finally, we would like to point out that the test already described in the manuscript is complementary to the one suggested here by the reviewer. The former evaluates whether NicheNet scores describing the potential influence of the 12 selected ligands on the set of inflammatory genes are significantly higher than on sets of randomly selected genes. The latter is a conventional over-representation analysis. In order to clarify this precise point in the main manuscript, we replaced this sentence:

“The top predicted target genes for these 12 ligands were enriched for inflammatory response gene sets (average p-value=3.25e-08 from Fisher’s exact tests after 10 cross-validation rounds).”

By this one:

“NicheNet scores describing the potential influence of the 12 selected ligands on the set of inflammatory genes are significantly higher than on sets of randomly selected genes (average p-value=3.25e-08 from Fisher’s exact tests after 10 cross-validation rounds)”

Minor points

It is not clear if this is included amongst the annotations already, but it would be good to have a very clear indication of the source of data, i.e. computationally inferred, manually annotated, complexes etc so that the user can filter with respect to quality. e.g. CellPhoneDB is much more accurate than the other resources, as it was manually curated and so provides better insight since it doesn't cloud the cell-cell networks with as much noise.

16. We thank the Reviewer for the suggestion, this would be indeed a useful feature. Implementing it all the way from pypath across the web service and OmnipathR is a major development, that we plan to do it in the close future and we opened a ticket in the bug tracker: <https://github.com/saezlab/pypath/issues/158>
Until this feature is deployed, we would like to note that some options are already available for filtering OmniPath data by the source and quality: for example, the network, the enzyme-PTM relationships and complexes can be filtered by either the `curation_effort` value (unique resource-reference pairs) or by the number of references. By removing the records with no references one can easily select the literature curated part of the data. In the *intercell* database a `consensus_score` is available for each *composite* record: the number of resources annotating the protein for a certain category (e.g. if three resources agree that a protein is a transporter that's more trustworthy compared to an annotation based on a single resource).

It would be nice to add some more information with respect to the definition of the complexes. At the moment all it says is "A complex is defined by its unique combination of members". Are all pull-downs e.g. considered complexes? Are complex components supposed to be universally present across multiple pull-down experiments? Some more information would be useful here.

17. Indeed this is a relevant point for anyone who works with protein complexes. However, this information is not available for all of the resources. To clarify this, we added the following sentence to the *Methods* section: *"We defined the complexes by their unique combination of members regardless of how the original resource processed the underlying experimental data."* In addition, we looked up the available details in the webpages and publications of the protein complex resources and added these to the <https://omnipathdb.org/info> page, see for example <https://omnipathdb.org/info#CORUM>.

I found the legends boxes in figures 2a,2b,2d and 3d confusing.

18. To improve clarity, we changed the position of the legend boxes and increased the spacing between the panels.

Reviewer 3

Integrated intra- and intercellular signaling knowledge for multicellular omics analysis
Dénes Túrei, Alberto Valdeolivas, Lejla Gul, Nicolás Palacio-Escat, Olga Ivanova, Attila Gábor, Dezső Módos, Tamás Korcsmáros, Julio Saez-Rodriguez*
MSB (Sept, 2020)

Summary

The authors continue to build on their previous work, developing OmniPath as a flexible and useful tool to capture molecular interactions from a wide-range of publicly available databases (Túrei et. al 2016). In this manuscript, OmniPath has been updated to incorporate integration of networks of inter- and intracellular signaling molecules. OmniPath was used to generate molecular inputs for cell-cell communication modeling for the two case studies, in concert with other analytical software, by combining information from numerous databases. They demonstrate its use with two case studies: 1) bulk RNA-seq data used to identify networks of differentially upregulated inflammatory ligands and their target genes of a Calu-3 lung epithelium cell line infected with SARS-Covid-2. They found proteins associated with the JAK-STAT and MAPK pathway pathways, which is supported by literature. 2) They used inter- and intracellular molecular interaction networks from single cell RNA-sequencing of 5 intestinal niche cells for ulcerative colitis, defining inflammatory signals underlying the diseased state. They suggested that the inflammatory signaling in ulcerative colitis derives from Treg and myofibroblast cell interactions, mediated by TLR pathways.

General remarks

The manuscript serves as an update to the capabilities of the OmniPath software, originally presented in Túrei et. al 2016. The case studies rely heavily on existing literature for source

data and validation for the conclusions drawn from their workflows that depend on their software. It is written for an audience that is familiar with cell-cell communication network modeling.

The authors do an excellent job of placing their software in the ecosystem of available molecular interaction databases, quantifying the coverage of each of their databases in comparison to others. They show a clear advantage to using their unifying tool.

The case studies provide the reader with examples of how OmniPath could fit into an analysis workflow, as well as that value of combining inter- and intracellular signaling interactions.

We thank the Reviewer for the positive comments.

Major Points

Overall, the main criticism is that the format of the manuscript does not clearly lay out use of the software, but instead emphasizes its place in the field's ecosystem of tools, which is important, but in the opinion of this reviewer, a secondary objective of a tool-based manuscript. A focus on a clear workflow with more detail for the case studies would make the software more accessible and highlight its importance in the field.

19. We agree that examples and workflows are greatly helpful for users of our tools. As the Reviewer suggested, we have restructured the manuscript. We have added a detailed workflow for the two case studies. Furthermore, we created a new set of tutorials with easily adaptable examples for some common tasks ("*Close connection to the analysis of omics data*" subsection of Results). We also implemented an integration of NicheNet in our OmnipathR Bioconductor package making the OmniPath -> NicheNet workflow (used in the first case study) very straightforward. Further changes on this direction are outlined in the responses to the comments below.

In general, the case studies serve as a demonstration of the software, but rely heavily on the results and interpretations of other published literature. To increase the impact of this paper, more work could be done to develop novel mechanisms using OmniPath. However, the emphasis of this manuscript is on tool development, so this is a secondary concern.

20. We agree with the Reviewer that since the focus is on the tool, the impact of the cases studies is secondary. That said, we have added some further insights to our case studies, in the Discussion to a very limited amount, and more in Appendix 2 and 2. We think this illustrates well the impact that dedicated application papers can achieve by using OmniPath.

Suggestions for improvement

- Well-written introduction. Can the authors include some exceptional example publications that use OmniPath, which can help expand the reader's appreciation for use-cases?

21. We thank the Reviewer for the good idea, we added the following sentence: *"It [OmniPath] has been used in many computational projects and omics studies. For example, to model cell senescence from phosphoarray data¹⁰, or as part of a computational pipeline to predict the effect of microbial proteins on human genes¹¹."*

- In the results section, briefly outline OmniPath's place in the tool ecosystem, but move major comparisons of OmniPath to other tools to supplementary (or last part of the results?).

22. We agree with the suggestion, we have restructured the Results and Discussion, and moved details, especially comparison to other tools, to Appendix 1. We also introduced Table 1 which defines OmniPath's profile compared to other resources.

- Put greater emphasis on each of the case studies (in light of the updates to the software - inter/intracellular signaling integration) and expand the methodology used in each with sufficient detail to guide the reader on its use and its integration with other software tools, from the perspective of a workflow (figures can illustrate and summarize this).

E.g. bring more of SNote 1 into the main text, develop a SNote 2 to support Case Study 2 as in 1, and bring some or all of SFig 6 into the main text. Also a diagram of the workflow of the case studies would be essential, this can serve as a template for future studies.

E.g. L273: Please go into greater detail describing the evidence for myofibroblast regulation of Treg pro-inflammatory responses to UC, highlighting key molecules and referencing STable 11.

23. In agreement with this comment and suggestions from the other reviewers, we extended the description of the case studies both in the main text and in Appendix 2. We now emphasise both the novel biological insights gained from these case studies (detailing the Treg cell related example more as suggested by the Reviewer) and the novel methodology. To support the latter one, we did a benchmark using the same Case Study 2 project but with other, well-known resources to point out the key novelties of using OmniPath (less false negative cell-cell interactions, less false positive intracellular interactions). Also, we created detailed computational workflows and tutorials which we publish online and refer from the Methods section.

- support the selection of candidate molecules (and other decisions) by reporting statistics

E.g. L248: How did the authors threshold their analysis for 12 ligands? Could they provide a justification? Were there any interesting down-regulated hits? Same for L273 candidates.

24. Here we followed the workflow of the NicheNet paper and applied a cut-off at similar proportions (selecting 12 out of 117 vs. 20 out of 131). As it is visible on Figure EV5b, this means the right tail of the density function of the Pearson correlation coefficients: to the left the distribution becomes steeper and lowering the threshold would include a much larger number of ligands. We rephrased the sentence about it in the Results section and added more details to Appendix 2: *"Out of a total of 117 overexpressed ligands, we selected the 12 top-ranked ones for subsequent analysis according to the distribution of correlation values (Figure EV5b) and nichenetr guidelines"*

All the ligands considered for prioritisation are overexpressed after SARS-CoV-2 infection, so we indeed found some interesting ligands not ranked among the top-12. For instance, *IL6* was not ranked among the top hits, but it has been proposed as a member of a molecular signature predicting severity and survival in COVID-19 patients (Del Valle et al. 2020¹). *TNF*, which is also included in the mentioned signature, was ranked second in our approach. Therefore, we were able to find some very relevant hits but also missed others. The method highly relies on the prior knowledge network and on the selected target genes. We created a new discussion section in the Appendix 2 where, among others, we commented on these topics.

Concerning the L273 candidates, the second case study has been largely extended in order to clarify this point (Results).

L225: As a demonstration and validation of OmniPath, these cases are useful and they agree with published literature. For both case studies, the inflammatory response gene ontology category is a very broad and expected one. Were there other gene sets that were less enriched but more descriptive or specific to the experimental condition indicating novel findings?

25. Indeed, the more specific categories might be more enriched, but we avoided these and chose the inflammatory response because it is well known to be a mainly intercellular communication driven process and also known to be relevant in SARS-CoV-2 infection. We give a more detailed answer for the similar question from Reviewer 1 (Answer 7). Also now we clarify this in the Discussion: *“Our study is limited to the relationship of autocrine signaling and inflammatory response, hence it doesn’t cover the complete process of viral infection.”*

In addition, we included a new discussion section in the Appendix 2 where we elaborate on the choice of the inflammatory response genes and consider the potential limitations of our approach.

Although the analysis was built from bulk data, is it possible for the authors to propose a map of cell-cell interactions based on the receptors and ligands involved in the described inflammatory response? For example, for the candidate chemokines, can they be mapped to putative leukocyte cell populations? e.g. IL23A to macrophages or dendritic cells, etc.

26. This is a very interesting point. As the reviewer mentions, the analysis was conducted on bulk data, not the most suitable type of data to infer maps of cellular communication. In addition, this is a sample with only lung epithelial cells, so that we can not study interactions with other cell-types such as leukocytes.

¹ Del Valle, D.M., Kim-Schulze, S., Huang, HH. et al. An inflammatory cytokine signature predicts COVID-19 severity and survival. Nat Med 26, 1636–1643 (2020). <https://doi.org/10.1038/s41591-020-1051-9>

If we had single cell RNAseq and/or Spatial transcriptomics data on a mixed population of e.g. epithelial or immune cells, we could have indeed performed an analysis as the one the reviewer suggests.

L256: The strategy and workflow in Case 2 are much clearer than Case 1. Perhaps a supplementary note could add some details (or add to the main text) about how the cell-cell interaction networks and disease-specific changes were modeled. As in SNote 1 for Case 1, the implications of these findings could be better explored as well.

27. The Reviewer is right, there are a number of details both regarding the method and the biological context which we couldn't include in the main text due to content limitations. We have extended and clarified Appendix 2 about the first case study, and largely rewrote the part about the second case study in the Results section and added more details.

Minor Points

L36: In the final paragraph of the introduction, it would be helpful to describe briefly the biological rationale behind the choices of the two case studies.

28. We added a sentence pointing on the main rationale, i.e. in these processes we can get more insight on the mechanism if cell-cell communication is considered: *Leveraging the intercellular communication knowledge in OmniPath we present two examples where autocrine and paracrine signaling are key parts of pathomechanism.*

L56: The authors mean 'some key players' instead of 'key players'?

29. Indeed, we added the word *some*

L144+: Can the authors introduce each database type by placing the value of its information into context of the field to better guide the reader in building their own workflows with the software?

30. We thank the Reviewer for the suggestion, we added such sentences to each subsection: About the network database: *"Interaction data is extensively used for a variety of purposes: for building mechanistic models, deriving pathway and TF activities from transcriptomics data and graph based analysis methods."* About the enzyme-PTM relationships: *"Enzyme-PTM relationships are essential for deriving networks from phosphoproteomics data or estimating kinase activities."* About the annotations: *"The annotations are helpful in omics data analysis, for example, can be used for contextualization or enrichment analysis."* About protein complexes: *"Many proteins operate in complexes, for example receptors often detect ligands in complexes. To facilitate analyses taking into consideration complexes, we added to OmniPath a comprehensive collection of 22,005 protein complexes..."*

L182: Fig 3b and 3c. Perhaps there is a better way to represent this information? It is difficult to distinguish categories at a glance with so many different colors. Perhaps with identifiable symbols? Are these colors appropriate for forms of color-blindness?

31. The alternative would be stacked bar plots, however those don't work with log scale. As the Reviewer recommended, we introduced a shape encoding to make sure all points are recognizable. We checked the colors by a color blindness simulation software and they are distinguishable with a few exceptions, and these are now addressed with the different marker shapes.

L196: For clarification, does the OmniPath protein complexes database contain information on states of the complex based on present constituent members, or be used to generate such information?

32. It is the former, a collection of protein complexes detected in experiments and contained in databases such as the Protein Data Bank or CORUM.

L228: Figure 4a is meant to summarize ways in which OmniPath can integrate with other software tools, but is unclear and difficult to interpret, particularly due to the layout of small arrows. This is in contrast to 4c, where the workflow is graphically clearer. As 4a is an important summary graphic, recommend redesign.

33. We thank the reviewer for noticing this, we completely redesigned this figure to give a more complete summary and improve clarity.

L251: Authors state in the text that Inflammatory ligands have downstream JAK-STAT targets - please report these targets, in the text and/or in Fig 4.

34. These are JAK2, STAT1, STAT3 and STAT4, we added them in the relevant sentence.

L253: The authors propose the JAK-STAT pathway could provide drug targets for Covid-19 treatment - can they expand on this, perhaps by leveraging their network-based approach, and propose candidate drug targets? Is it possible to rank drug targets, say based on limited off-target genes, or by connecting to DGIdb, etc. In SNote 1, Ruxolitinib was mentioned as a potential JAK-STAT-interacting drug, but this was already introduced by the authors of the source data (Blanco-Melo et al.).

35. We thank the reviewer for this relevant suggestion. We would like to mention that we do not propose that the JAK-STAT pathway could provide drug targets for COVID-19. We aim at highlighting that we found many genes involved in the JAK-STAT pathway. This pathway has been recurrently mentioned in the COVID-19-related literature as a potential target to treat SARS-CoV-2 infection, as in Blanco-Melo et al, hence supporting our OmniPath-based approach.

In order to dig a bit further into the biology of our results, we explored the drugs targeting the genes shown in Figure 4b (Dataset EV14). As expected, we found many compounds used in the treatment of multiple inflammatory diseases such as rheumatoid arthritis, inflammatory bowel disease and multiple sclerosis. Among the most interesting results, we identified minocycline, an antibiotic and anti-inflammatory drug targeting *CASP3*. *CASP3* is a marker of caspase-dependent apoptosis which interestingly shows an increased activity in the presence of the SARS-CoV-2-encoded protein ORF3a (Ren et al 2020²). Minocycline has been very recently proposed to alleviate the effects of SARS-CoV-2 severe infection in the central nervous system (Oliveira et al., 2020³). In addition, minocycline successfully decreases inflammatory cytokines such as *TNF*, which is highly expressed in severe COVID19-patients and linked to an increased neurological damage (Sharma et al., 2018⁴, Chen et al., 2020⁵). It is to note that we also identified *TNF* as a top ligand influencing immune response in our approach.

We included these results in the Appendix 2 and Dataset EV14. In addition, we included the following sentences in the main text of the manuscript:

“To further characterize the potential medical relevance of these results, we investigated the drugs targeting the genes shown in Figure 4b (Dataset EV14). Among the most interesting results, we identified minocycline, an antibiotic and anti-inflammatory drug targeting CASP3 and TNF. Minocycline has been very recently proposed to alleviate the effects of SARS-CoV-2 severe infection in the central nervous system³”

L254: The authors reported true-positive results about inflammatory pathways. Would they be able to argue if there are other important pathways reported to be promoted by SARS-CoV-2 infection that has not been captured here? They can argue the reason in the discussion section.

36. We completely agree about the importance of clearly speaking about the limitations of these case studies. Many other pathways and biological processes are perturbed by the SARS-CoV-2 infection, such as fatty acid metabolism (Figure EV6a), but their link to cell communication is not so straightforward. We rephrased the Discussion to emphasize that the scope of the study was

2 Ren, Y., Shu, T., Wu, D. et al. The ORF3a protein of SARS-CoV-2 induces apoptosis in cells. *Cell Mol Immunol* 17, 881–883 (2020). <https://doi.org/10.1038/s41423-020-0485-9>

3 Oliveira AC, Richards EM, Karas MM, Pepine CJ and Raizada MK (2020) Would Repurposing Minocycline Alleviate Neurologic Manifestations of COVID-19? *Front. Neurosci.* 14:577780. doi: 10.3389/fnins.2020.577780

4 Sharma, R. K., Oliveira, A. C., Kim, S., Rigatto, K., Zubcevic, J., Rathinasabapathy, A., et al. (2018). Involvement of neuroinflammation in the pathogenesis of monocrotaline-induced pulmonary hypertension. *Hypertension* 71, 1156–1163. doi: 10.1161/HYPERTENSIONAHA.118.10934

5 Chen, N., Zhou, M., Dong, X., Qu, J., Gong, F., Han, Y., et al. (2020). Epidemiological and clinical characteristics of 99 cases of 2019 novel coronavirus pneumonia in Wuhan, China: a descriptive study. *Lancet* 395, 507–513. doi: 10.1016/S0140-6736(20)30211-7

only autocrine signaling and inflammatory response: *"The first case study pointed to potential signaling mechanisms of autocrine origin in SARS-CoV-2 infection which can contribute to the dysregulated inflammatory and immune response characteristic of severe COVID cases. Our study is limited to the relationship of autocrine signaling and inflammatory response, hence it doesn't cover the complete process of viral infection."*

In addition, we created a new discussion section in the Appendix 2 to elaborate on our choices and discuss the potential limitations of our approach.

L268: Missing the word "cells" after "Treg".

37. Corrected

L273: "TLR431 and TLR3 pathways³² were upregulated in UC." In Treg cells?

38. We added *"In contrast, also in Treg cells, ..."*

L288+ (Discussion section): Can the authors suggest best practices that would improve the addition of new resources to future iterations of OmniPath? How can database-derived interaction networks such as those built with OmniPath start to incorporate magnitude of cell signaling events?

39. Database building is a subdiscipline within computational biology on its own. We are very grateful for the fantastic work of all the colleagues who created the resources contributing to OmniPath. After processing more than a hundred of databases definitely we have some opinion about best practices and ways to improve this field. If we wanted to write about these that would be enough for another paper. For this reason we feel this topic out of scope for the current manuscript.

Consider suggesting how to extend case study results by experimentation, or other opportunities to build on this work. Spatial annotation suggested by the authors is an excellent example. Can the authors expand further?

40. We agree with the Reviewer that suggesting how the *in silico* work done with OmniPath can be validated and experimentally analysed is a meaningful addition to the manuscript. Therefore, we extended the Discussion section on how each case study can contribute to experimental design and added the type of experiments which could complement and validate the analyses generated with OmniPath.

We also elaborated in the discussion the impact of OmniPath on the analysis of spatial data.

For further discussion, what major challenges remain, either for the authors, the researchers generating the resources, or the end-users?

41. This is indeed an important question, we added a new paragraph to the Discussion. *“Over the past four years we have kept developing OmniPath, adding new features and resources every couple of months. One of our main objectives for the future is to add more context information e.g. cell type and physiological condition to the signaling network, and use scores to prioritize interactions and paths which contribute stronger to indirect causal relationships. Towards these aims we plan to leverage text mining methods such as INDRA⁴². We are also working on benchmarking the intercellular communication knowledge in OmniPath and in alternative resources by deriving ground truth from experimental data⁴⁶. Furthermore, we would like to extend OmniPath with pathogen-host interactions²⁸.”*

L622: The Blanco-Melo, D. 2020 Microbiology reference from the main text (Reference 23) SNote 1 (Reference 1) seems to be incorrect. It was originally published under that title in BioRxiv, but later peer-reviewed and published under a new title in Cell here: doi: 10.1016/j.cell.2020.04.026

42. We updated the reference to point to the Cell paper both in the main text and in Appendix 2.

L730: How do you interpret and threshold NicheNet's ligand-target regulatory potential and ligand-receptor interaction potential statistics in SFig6c,d? How do these statistics compare to ligand-target/ligand-receptor statistics for those not chosen? Clarification of these in the text would strengthen the interpretation of the analysis for the reader.

43. We thank the reviewer for pointing out these relevant points. In our manuscript, we did not want to overextend these points since they are detailed in the NicheNet original publication.

Regarding the selected threshold, we provide a detailed answer above (Answer 24).

Regarding the ligand-target regulatory potential and ligand-receptor interaction potential scores. NicheNet is based on the idea of propagation of a signal in a network, from a ligand to its receptor(s), downstream signaling proteins, transcription factors and finally to the targets of these TFs. They simulated this process by applying PageRank, a random-walk based method. The PageRank method provides a score indicating the likelihood to reach a downstream protein when the propagation starts from a given ligand. NicheNet's ligand-target regulatory potential and ligand-receptor interaction potential scores are based on the PageRank results. We tried to clarify this point in the methods section:

“Then, we computed ligand–target regulatory potential scores based on the topology of our aforementioned networks, following the protocols described in the NicheNet original study and

using its associated nichenetr package. Briefly, NicheNet is based on the idea of propagation of a signal in a network, from a ligand to its receptor(s), downstream signaling proteins, transcription factors and finally to their regulatory targets. NicheNet simulates this process by applying PageRank, a random-walk based method. The PageRank method provides a score indicating the likelihood to reach a downstream protein when the propagation starts from a given ligand. Finally, to provide a score estimating the potential influence of every ligand in the expression of target genes, a matrix containing the PageRank scores is multiplied by the weighted adjacency matrix of the gene regulatory network.”

44. Regarding the comparison of ligand-target scores for inflammatory response with ligand-target scores of non-chosen targets, we already evaluated to which extent our top 12 prioritized ligands can together predict whether the top predicted targets belong to the inflammatory response gene set or not. This is already detailed in the main manuscript and Appendix 2:

“The top predicted target genes for these 12 ligands were enriched for inflammatory response gene sets (average p-value=3.25e-08 from Fisher’s exact tests after 10 cross-validation rounds).“

In order to further clarify this point we replaced this sentence by the following:

“NicheNet scores describing the potential influence of the 12 selected ligands on the set of inflammatory genes are significantly higher than on sets of randomly selected genes (average p-value=3.25e-08 from Fisher’s exact tests after 10 cross-validation rounds)”

Can the authors report statistics on conflicting annotations from different databases as part of their comparison to other tools and molecular coverage, perhaps in the supplementary material?

45. We created a new figure (Appendix Figure S4) which presents certain potential inconsistencies in the network causality and the intercellular annotations. This is interesting information, however we need to note, such inconsistencies might result either from a mistake of the database authors or from biological reality.

RE: MSB-20-9923R, Integrated intra- and intercellular signaling knowledge for multicellular omics analysis

Thank you again for sending us your revised manuscript. We have now heard back from the three reviewers who were asked to evaluate your study. As you will see below, the reviewers think that the study has improved after the performed revisions and they are supportive of publication. Reviewer #1 recommends a couple of edits, which we would ask you to perform in a minor revision.

Moreover, we would ask you to address a few remaining editorial issues listed below.

Reviewer #1:

The authors did a good job in answering most of my original comments, and overall I think the revised version better illustrates the capabilities and contributions of the work. They provide more comparisons to similar resources (e.g. Pathway Commons), better describe the two case studies, and added a number of workflows to facilitate integration of OmniPath with other tools.

Although the authors did not include a direct comparison between NicheNet with the original network, and NicheNet with Omnipath as I suggested, I think the added workflows would allow users now to do these kind of comparisons.

Regarding extending the case studies and highlighting more biological novelty, I think the revision is quite limited. Although the authors discuss more in depth the case studies, I did not see a lot of additional biological insights. The authors only discuss a bit more potential drug targets and add some more references, but I think it would be nice if the description of the results could be made more specific. For example for case study 2, it would be nice if the authors could include a visualisation of the ligand-receptor network between myofibroblasts and Tregs in Ulcerative Colitis. In this way, the reader can get an idea about the specific ligand-receptor interactions, and not only about the downstream affected pathways, as is showcased now. For the visualization of the interactions, it would then be easy to show which ones are also found through other resources, and which ones not, instead of just comparing the number of found ligands/receptors as is done now. I think this would not be too much added work, and would greatly add value to a more biologically oriented reader of the manuscript.

Overall when reading the revised manuscript again, I still noted quite a number of spelling mistakes (e.g. line 273-274, line 285, line 560, undefined reference (j) in the legend of Table 1, etc.).

Reviewer #2:

The authors have adequately addressed my comments and concerns and with the addition of multiple other resources, workflows, implementation of integration with NicheNet and many other features I believe that it will be a fantastic resource for the community and widely used. The case studies are well supported and the comparison that they have provided for UC shows that the integration of the resources provides more detailed information. All the case studies are also now better described and annotated and better showcase the value of this resource. The only comment I still have is regarding the complexes and I think there I am confused: when the authors say "A complex is defined by its unique combination of members" do they mean "A complex is defined by its combination of unique members"? If so it would be good to amend this in the two places where it appears.

Reviewer #3:

I really like where this revision ended up. The authors have done a good job of addressing my concerns and the paper is now suitable for publication, in my opinion.

We were glad to receive the largely positive response from you and the reviewers to the revised version of our manuscript "Integrated intra- and intercellular signaling knowledge for multicellular omics analysis". Hereby we submit an updated version of our manuscript where we address all points in your letter.

- Following the suggestions from Reviewer #1, we created a new figure (Figure 5c) with a visualization of condition specific myfibroblast-Treg cell ligand-receptor interactions in healthy and ulcerative colitis conditions.
- Due to the introduction of this new figure panel, the former Figure 4a became Figure 4, and former Figure 4b and 4c became Figure 5a and 5b, respectively
- We added a couple of new sentences to the Results (page 14) and Methods (page 26) about the new figure; the former also adds more details about the biological interpretation of the second case study
- Reviewer #1 also suggested that in this figure we could "show which ones are also found through other resources, and which ones not". This would be only possible by replicating the figure for each resource we want to show. Apart from the arbitrary decision which resources to include, in our opinion it wouldn't add much useful information over the already provided quantitative comparison.
- We corrected many typos, language and formatting errors, including the ones Reviewer #1 kindly collected

We really liked the suggestion from Reviewer #1, we believe it became a great addition to the manuscript. We hope with the updates the manuscript will be acceptable for publication in Molecular Systems Biology, please let us know if we should do any further changes.

RE: MSB-20-9923RR, Integrated intra- and intercellular signaling knowledge for multicellular omics analysis

Thank you again for sending us your revised manuscript. We are now satisfied with the modifications made and I am pleased to inform you that your paper has been accepted for publication.

Corresponding Author Name: Julio Saez-Rodriguez

Manuscript Number: MSB-20-9923R